# Boundary conditions investigation to improve computer simulation of cerebrospinal fluid dynamics in hydrocephalus patients

Seifollah Gholampour [1✉] & Nasser Fatouraee[2]

Three-D head geometrical models of eight healthy subjects and 11 hydrocephalus patients were built using their CINE phase-contrast MRI data and used for computer simulations under three different inlet/outlet boundary conditions (BCs). The maximum cerebrospinal fluid (CSF) pressure and the ventricular system volume were more effective and accurate than the other parameters in evaluating the patients' conditions. In constant CSF pressure, the computational patient models were 18.5% more sensitive to CSF volume changes in the ventricular system under BC "C". Pulsatile CSF flow rate diagrams were used for inlet and outlet BCs of BC "C". BC "C" was suggested to evaluate the intracranial compliance of the hydrocephalus patients. The results suggested using the computational fluid dynamic (CFD) method and the fully coupled fluid-structure interaction (FSI) method for the CSF dynamic analysis in patients with external and internal hydrocephalus, respectively.

[1] Department of Biomedical Engineering, North Tehran Branch, Islamic Azad University, Tehran, Iran. [2] Biological Fluid Mechanics Research Laboratory, Biomechanics Department, Biomedical Engineering Faculty, Amirkabir University of Technology, Tehran, Iran. ✉email: s.gholampour@iau-tnb.ac.ir

Cerebrospinal fluid (CSF) acts as a medium for transporting nutrients and neuroendocrine substances and removing toxic metabolites. It also preserves the brain's chemical environment[1] and protects the central nervous system (CNS) from a mechanical point of view[2]. There are diseases like hydrocephalus syndrome, in which the intracranial pressure (ICP) increases due to the same cause that has led to the disease[3]. Hydrocephalus is caused by abnormal accumulation (imbalance between production and circulation) of CSF within the brain[3]. Hydrocephalus is still considered clinically complex since it has unknown pathophysiology aspects and various management methods[4]. Thus, more comprehensive studies are required to gain more in-depth insight into the nature of the disease.

Experimental methods have limitations in measuring parameters affecting the disease[5,6]. They are also insufficiently accurate in measuring biomechanical loadings and local flow patterns[7]. Thus, many studies have been carried out using computerized biomechanical simulation. The hydrodynamic simulation of temporal and spatial CSF flow-distribution patterns may raise the awareness of hydrocephalus complexities and their practical clinical value. However, the most challenging part of biomechanical simulations is boundary conditions (BCs)[8,9]. Results calculated with solution methods such as fluid-structure interaction (FSI) are highly sensitive to BCs[10,11]. Generally, the study of BCs in hydrocephalus modeling includes evaluating changes in both essential BCs (e.g., non-slip-boundary conditions) and natural BCs (e.g., pressure/load). This study mainly aimed to evaluate the effects of natural BC changes on the computational biomechanics of hydrocephalus.

In the past, various computational methods and BCs were used to analyze hydrocephalus (Table 1). Taylor et al. carried out a 2D head simulation for a hydrocephalus patient[12]. The finite-element method (FEM) with CSF pressure as BC was used in that study. The ventricular system and brain tissue have many geometrical complexities, affecting problem-solving conditions considerably. Hence, the 2D simulation cannot fulfill the requirements regarding the geometrical complexities. Wirth et al. and Cheng et al. analyzed the biomechanics of hydrocephalus using 3D FEM modeling[13,14]. Although 3D studies better show the geometrical complexities than 2D studies, the FEM solution can analyze solid models. However, it cannot simulate the CSF flow circulation in the head[15]. It should also be mentioned that the first 3D FEM models in the literature were simplified[13], which did not reflect the real geometrical complexities of the head. Therefore, to remedy this shortcoming, some previous studies concentrated on the computational fluid dynamic (CFD) simulation of the CSF flow with various inlet/outlet BCs.

Jacobson et al. analyzed the fluid flow in the cerebral aqueduct (CA) of a hydrocephalus patient with constant and sinusoidal pressures, and velocity functions as inlet BC, and zero pressure as the outlet BC[16,17]. Hadzri et al. modeled the third and fourth ventricles of a healthy subject and a hydrocephalus patient with the pulsatile CSF flow rate and constant pressure as the inlet and outlet BCs, respectively[18]. Fin et al. simulated the CSF flow in the CA of a healthy subject and used pulsatile velocity as the inlet BC[19]. Howden et al. modeled CSF dynamics in a healthy subject with pulsatile flow rate as the inlet BC and constant pressure in the outlet of ventricles as the outlet BC[20]. Kurtcuoglu et al. set

**Table 1 The models and boundary conditions of previous computerized biomechanical simulations[12,14,16-34].**

| Authors | Solving method | Boundary conditions | Reference |
|---|---|---|---|
| Taylor et al. | 2D FEM | External loading: 3000 Pa | 12 |
| Cheng et al. | 3D FEM | Pressure gradient between the brain and ventricles: 200 Pa | 14 |
| Jacobson et al. | 3D CFD (simplified model) | (1) Inlet: constant pressure<br>Outlet: zero pressure<br>(2) Inlet: constant velocity<br>Outlet: zero pressure<br>(3) Inlet: $1 + 2 \sin(2\pi t)$ Pa<br>Outlet: zero pressure | 16 |
| Jacobson et al. | 3D CFD (simplified model) | Inlet: $80 + 160 \sin(2\pi t)$ Pa<br>Outlet: zero pressure | 17 |
| Hadzri et al. | 3D CFD | Inlet: pulsatile flow rate<br>Outlet: 500 Pa | 18 |
| Fin et al. | 3D CFD (simplified model) | Parabolic inlet velocity is considered as BC. | 19 |
| Howden et al. | CFD | Inlet: pulsatile flow rate<br>Outlet: constant pressure | 20 |
| Kurtcuoglu et al. | 3D CFD (simplified model) | Inlet: oscillatory wall motion<br>Outlet: zero pressure | 21 |
| Kurtcuoglu et al. | 3D CFD | Inlet: transient velocity<br>Outlet: zero pressure | 22,23 |
| Gupta et al. | 3D CFD | Inlet: pulsatile velocity<br>Outlet: zero pressure | 24,25 |
| Farnoush et al. | 3D CFD | Inlet: pulsatile velocity<br>Outlet: zero pressure | 26 |
| Linninger et al. | 3D CFD | Inlet: pulsatile flow<br>Outlet: constant pressure | 27 |
| Linninger et al. | 3D FSI | Inlet: pulsatile flow<br>Outlet: 500 Pa (healthy subject) and 2700 Pa (hydrocephalus patient) | 28 |
| Gholampour et al. | 3D FSI | Inlet: pulsatile flow<br>Outlet: constant pressures | 29 |
| Sweetman et al. and Gholampour et al. | 3D FSI | Inlet: pulsatile flow<br>Outlet: 500 Pa (healthy subject) and 2700 Pa (hydrocephalus patient) | 30,31 |
| Gholampour et al. | 3D FSI | The flow rate diagrams are considered as BCs | 32-34 |

*BC* boundary condition, *FEM* finite element method, *CFD* computational fluid dynamic, *FSI* fluid-structure interaction.

zero pressure as the outlet BC to assess CSF hydrodynamics in healthy subjects and hydrocephalus patients[21]. In other studies, Kurtcuoglu et al. considered the effect of brain motion by specifying the explicit boundary grid motion in the third ventricle and CA[22,23]. Gupta et al. modeled the subject-specific CSF flow and defined the pulsatile velocity field and zero pressure as the inlet and outlet BCs[24,25]. Farnoush et al. studied the effect of endoscopic third ventriculostomy on treating a hydrocephalus patient using the biomechanical simulation of CSF hydrodynamics in the third and fourth ventricles[26]. They used pulsatile velocity and zero pressure as the inlet and outlet BCs, respectively. However, the pure CFD method fails to reflect the complexities of the fluid–solid boundary in the computer simulation of hydrocephalus[27]. The brain's inner and outer surfaces, which are in contact with CSF, deform due to volume changes of the brain and ventricles, resulting from the disease. However, the CFD method fails to consider deformable boundaries. Hence, some other studies were dedicated to FSI simulation.

Linninger et al., Sweetman et al., and Gholampour et al. performed some studies by the FSI simulation of the CSF flow and brain tissue in healthy subjects and hydrocephalus patients[28–32]. They defined the CSF flow as the inlet BC and different constant pressures in the healthy subjects and patients' sagittal sinus as the outlet BC. In other studies, Gholampour et al. examined the correlation between CSF hydrodynamic changes and hydrocephalus patients' clinical symptoms before and after shunting with the flow rate as BCs[32–34]. In this study, the effect of different combinations of BCs was quantitatively evaluated on changes in hydrodynamic parameters of the CSF flow to find the most accurate BCs for simulating hydrocephalus patients.

## Results
Results of FSI simulations were calculated using ADINA software in five cardiac cycles. As there was no apparent difference between the data of 4th and 5th cycles, the last cycle results are reported. The CSF flow dynamics in the healthy subjects and patients (after contracting disease) were calculated and compared under three BCs.

**Evaluation of CSF flow dynamics**. The fluid-flow function is of great significance for examining neurological diseases[35] such as hydrocephalus. Hence, in the first section, the CSF flow-rate diagram was evaluated at CA. The difference between the positive areas above the x axis and the negative areas under the x axis in the CSF flow-rate diagram reflects the CSF stroke volume, which is the volume of CSF passing into the aqueduct in diastole and systole phases[36]. The aqueductal CSF stroke volume is useful for assessing hydrocephalus patients' treatment process and is used as a shunt-responder predictor in patients treated with shunting[37]. This parameter is also an index for assessing the degree of irreversibility of neuronal damages due to hydrocephalus disease during the treatment process[5]. Hence, the aqueductal CSF stroke volume was investigated in this study due to its significance. The difference in the mean values of the maximum aqueductal CSF stroke volume between BCs "A", "B", and "C" was about 0.9% in the group of patients and about 0.3% in the group of healthy subjects (Fig. 1a and Supplementary Data 1). The standard error (SE), standard deviation (SD), and coefficient of variation (CV) values of the maximum aqueductal CSF stroke volume data under all the BCs were minimal in both groups of patients and healthy subjects. These findings, along with the results of the analysis of variance (ANOVA) and Student's t tests, indicated that the maximum aqueductal CSF stroke volume data had an acceptable range of dispersion for both groups. The results, based on BCs "A", "B", and "C", showed that

the mean value of the maximum aqueductal CSF stroke volume in the group of patients was about 7.9 times that in the group of healthy subjects (Fig. 1a and Supplementary Data 1).

In the second section, the Reynolds number was calculated. The Reynolds number (Re = $\rho_F uD/\mu$; D is the aqueduct diameter) is a nondimensional parameter for evaluating the laminarity or turbulence of flow[38]. The Reynolds number is the ratio of inertial forces to viscous forces[39]. The results showed that the mean value of the maximum Reynolds numbers of the CSF flow in the patients and healthy subjects was $388.6 \pm 6.2$ and $332.3 \pm 5.1$ under BC "A", $374.3 \pm 5.9$ and $323.3 \pm 5.0$ under BC "B", and $372.4 \pm 5.8$ and $321.8 \pm 5.0$ under BC "C", respectively. The results also demonstrated that the difference of the mean values of the maximum Reynolds number between BCs "A", "B", and "C" was <4.4% in the group of patients and <3.3% in the group of healthy subjects. It is noteworthy that CV for the maximum Reynolds number under all the BCs was <4.6% in the group of patients and less than 4.5% in the group of healthy subjects. Based on the results of ANOVA and Student's t tests and the values of SE, SD, and CV, the data dispersion of the maximum Reynolds number was in an acceptable range in both groups. According to the results, the mean value of the maximum Reynolds numbers in patients under BCs "A", "B", and "C" was respectively 16.9%, 15.8%, and 15.7% higher than that in the healthy subjects.

The CSF pressure diagrams in the subarachnoid space (SAS) and CA of patient No. 7 under BCs "A", "B", and "C" are compared in Fig. 1b and three snapshots of CSF pressure distribution under BC "B" in the patient No. 7 are shown in Fig. 1c–e. Investigating the pressure diagrams of the healthy subjects and patients revealed that in all the healthy subjects and patients, the maximum pressure occurred at 84% of the cardiac cycle, i.e., during the early systole, while the minimum pressure occurred at 17.5% of the cardiac cycle, i.e., during mid-systole. These results agree with those of previous studies[28–30,32]. On the other hand, the maximum velocity in a cardiac cycle occurred at 17.5% of the cardiac cycle (Fig. 1f, g). This result agrees with the results of an experimental study by Ünal et al. and a numerical study by Gholampour et al.[30,32,38,40]. CSF velocity distribution at 17.5% (mid-systole) of the cardiac cycle is compared between all the patients under BC "B" in Fig. 2.

As the absolute value of ICP is not directly measurable using MRI, the analysis of intracranial dynamics cannot be performed entirely based on imaging. While having information on the absolute ICP value could be essential for assessing patients' clinical conditions, it is impossible to calculate the absolute value of ICP by computational fluid mechanics, as only the pressure gradient appears in fluid motion equations. In biphasic models, pressure acts as the fluid phase's pressure, but its level is undetermined. Hence, we did not calculate absolute ICP. In this study, as in many previous studies, CSF pressure calculated by biomechanical simulation in the pathway of CSF circulation is called CSF pressure[26,30,32,33], and CSF pressure in the upper convexity of the brain in SAS and CA is called CSF pressure in SAS and CSF pressure in CA, respectively. Based on the pulsatility of pressure and to achieve numerical values reported in this paper, first, the CSF pressure diagram based on the cardiac cycle was calculated for the healthy subjects and patients at each location, such as the upper convexity of the brain in SAS and CA (Fig. 1b). Then, the peak values of these diagrams were reported as CSF pressure in SAS and CA, respectively, in Tables 2 and 3.

According to the results of Table 2, the mean values of the maximum CSF pressure in the SAS and CA of the healthy subjects under BC "A" were $650.0 \pm 22.1$ and $623.0 \pm 21.9$ Pa, respectively. The corresponding data were, respectively, $586.2 \pm 20.5$ and $561.5 \pm 19.2$ Pa for BC "B" and $493.2 \pm 8.1$ and $484.1 \pm 8.0$ Pa for BC "C" (Table 2). According to Table 3, the mean

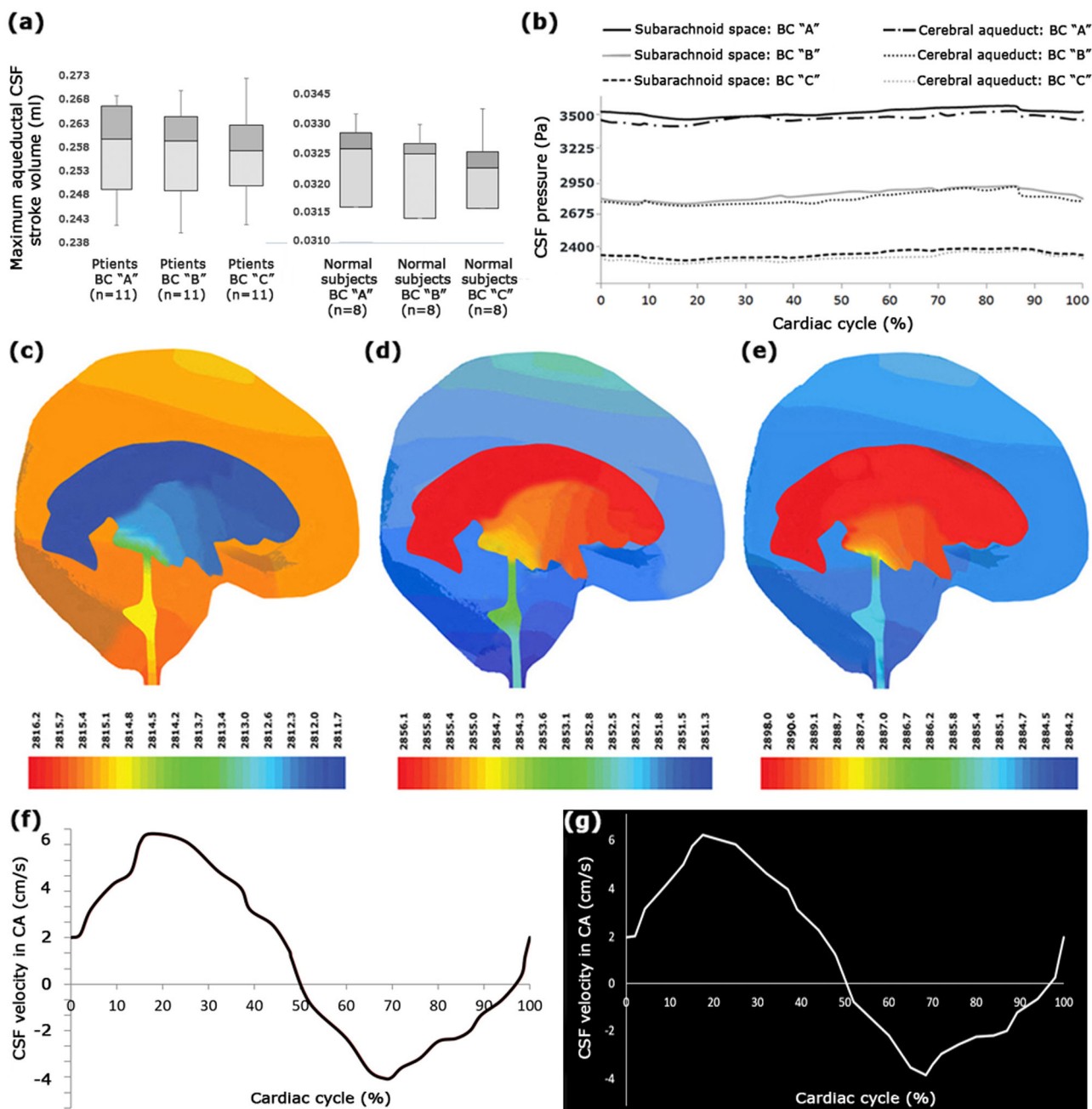

**Fig. 1 Maximum aqueductal stroke volume, pressure, and velocity of CSF flow. a** The values of the maximum aqueductal CSF stroke volume for healthy subjects and patients under BCs "A", "B", and "C". It should be noted that there were 8 healthy subjects and 11 hydrocephalus patients in this study. **b** compares the CSF pressure diagrams in the SAS and CA of patient No. 7 under BCs "A", "B", and "C". **c–e** display three snapshots of CSF pressure distribution under BC "B" in the patient No. 7 at 17.5% (mid-systole), 64% (diastole), and 84% (early systole) of the cardiac cycle, respectively. The units of the color scale are Pascal. Panels (**f**) and (**g**), respectively, show the CSF velocity diagram calculated with FSI simulation and the in vivo-measured diagram based on the cardiac cycle under BC "C" in the CA of patient No. 7. Raw data for Fig. 1a are included in Supplementary Data 1. CV coefficient of variation, SE standard error, BC boundary condition, CSF cerebrospinal fluid, CA cerebral aqueduct.

values of the maximum CSF pressure in the SAS and CA of patients under BC "A" were 3592.3 ± 103.7 and 3568.9 ± 105.1 Pa, respectively. The corresponding data were 3009.9 ± 85.8 and 2984.8 ± 86.6 Pa for BC "B" and 2516.5 ± 35.3 and 2503.5 ± 35.0 Pa for BC "C" (Table 3).

In contrast to maximum values of velocity, aqueductal stroke volume, and the Reynolds number, the calculation of pressure in patients under BCs "A", "B", and "C" revealed a significant difference (>42%) between the mean values of maximum pressure under three BCs. The CVs of the maximum CSF pressure in SAS

for BCs "A", "B", and "C" were 9.6%, 9.5%, and 4.7%, respectively. The corresponding data for the maximum CSF pressure in CA were 9.8%, 9.6%, and 4.6%, respectively. As observed, the CVs of the maximum CSF pressure under BCs "A" and "B" were more than twice the CV of the maximum CSF pressure under BC "C".

Finally, the results showed that the mean values of the maximum CSF pressure in SAS in the group of patients under BCs "A", "B", and "C" were 5.5, 5.1, and 5.1 times the similar values in the group of healthy subjects, respectively.

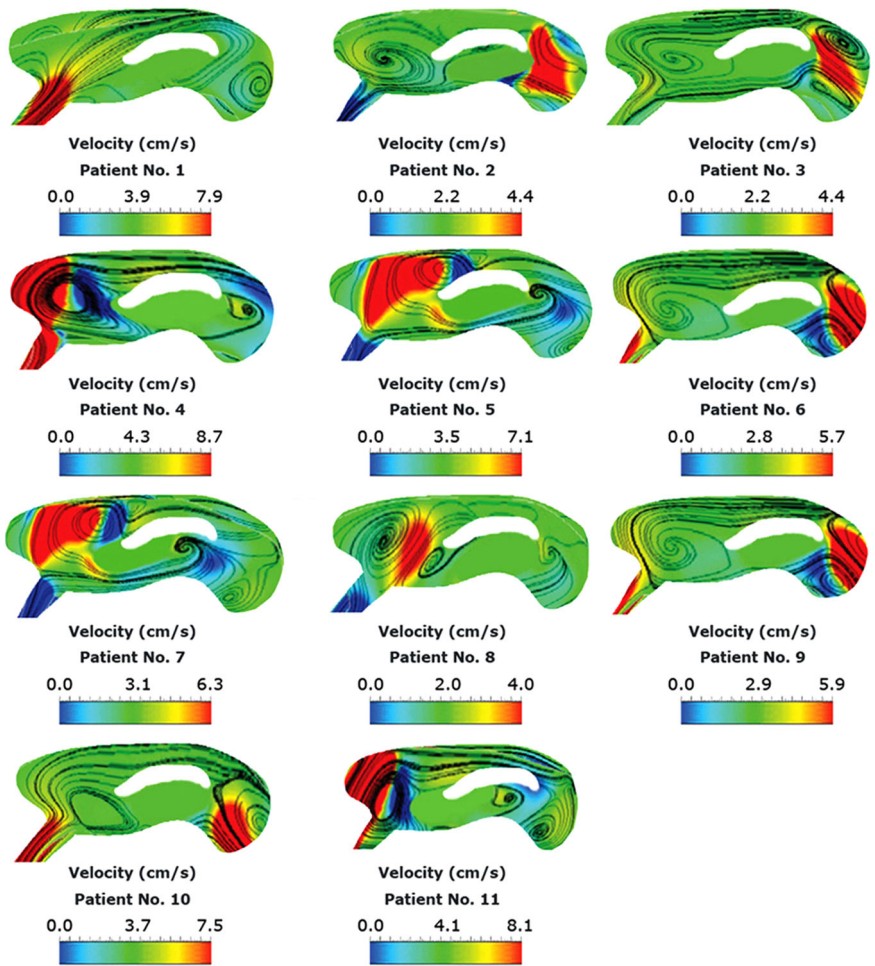

**Fig. 2 Distribution of CSF velocity in patients.** Panels show CSF velocity distribution at 17.5% (mid-systole) of the cardiac cycle in all the patients under BC "B". BC boundary condition, CSF cerebrospinal fluid.

The corresponding data for the maximum CSF pressure in CA were 5.7, 5.3, and 5.2, respectively (Tables 2 and 3).

**Comparison of experimental and biomechanical simulation data.** First, the CSF velocity diagram in CA was calculated from the FSI simulation of each healthy subject, and each patient was compared with the CSF velocity diagram in CA, which was measured experimentally using the CINE phase-contrast magnetic resonance imaging (CINE PC-MRI) of them. The reason was that CA had the smallest cross-sectional area in the CSF circulation pathway, and thus, CSF fluid had the most significant velocity in this section based on the law of continuity in fluid mechanics. The computational results provided the velocity data in mesh nodes, whose locations matched the pixel locations in CINE PC-MRI CSF velocity measurement. The CSF velocity diagrams calculated by FSI simulation and the experimentally measured diagram based on the cardiac cycle under BC "B" in the CA of patient No. 7 are shown in Fig. 1f and g, respectively. Due to the importance of the maximum CSF velocity in fluid-flow conditions, the maximum CSF velocity values in the CA of all healthy subjects and patients under the three BCs were obtained using CINE PC-MRI and FSI simulation and compared according to Fig. 3a–c and Supplementary Data 2. The $P$ values in the Shapiro–Wilk test for all velocity datasets were >0.78, which confirmed the normality of all experimental and FSI simulation data of CSF velocities. The $P$ values in all Student's $t$ tests were >0.76. It should be noted that in all Student's $t$ tests, the $P$ value of variance equality was >0.64. Moreover, the results of Fig. 3a–c and Supplementary Data 2 demonstrated that the boundary conditions fall in the 95% confidence interval of the CSF velocity data.

Maximum differences between the maximum CSF velocity in CA measured with CINE PC-MRI and the maximum CSF velocity calculated using FSI simulation under BCs "A", "B", and "C" for all healthy subjects and patients were less than 4.8%, 3.6%, and 2.3%, respectively (Fig. 3a–c and Supplementary Data 2). The maximum phase difference between the experimentally measured and computer-simulated velocity diagrams under all the BCs and in all healthy subjects and patients was <0.9% (Fig. 1f, g). Although the CSF velocity results based on BC "C" were closer to the CINE PC-MRI data, differences between the maximum CSF velocities of the three BCs were <3.3%.

Second, the SAS's CSF pressure values were experimentally measured in 10 of the 11 patients by continuous monitoring of ICP using a reliable sensor that introduced 1–2 cm through a minimal opening and a small burr hole[41]. CSF pressure in SAS was not measured in healthy subjects due to the invasiveness of ICP monitoring and the subjects' health. Moreover, CSF pressure in SAS was not measured in patient No. 11 because of his special conditions and limitations. The $P$ values in the Shapiro–Wilk test for all CSF pressure datasets were >0.66, which confirmed the normality of all CSF pressure data of ICP monitoring and FSI simulation. The $P$ values in all Student's $t$ tests were >0.65.

**Table 2 The maximum CSF pressure details and volumes of head substructures of the eight healthy subjects.**

| Healthy Subjects / Variables | No. 1 | No. 2 | No. 3 | No. 4 | No. 5 | No. 6 | No. 7 | No. 8 | Mean | Standard deviation | Standard error | Coefficient of variation (%) | 95% Confidence interval Lower | Upper |
|---|---|---|---|---|---|---|---|---|---|---|---|---|---|---|
| Maximum CSF pressure in SAS under BC "A" (Pa) | 599.6 | 699.1 | 719.1 | 704.2 | 589.3 | 594.3 | 710 | 584.3 | 650.0 | 62.5 | 22.1 | 9.6 | 597.7 | 702.3 |
| Maximum CSF pressure in SAS under BC "B" (Pa) | 575.2 | 622.8 | 655.1 | 614.1 | 515.6 | 570.2 | 641.5 | 495.1 | 586.2 | 58.0 | 20.5 | 9.9 | 537.7 | 634.7 |
| Maximum CSF pressure in SAS under BC "C" (Pa) | 505.4 | 508.5 | 515.9 | 499.1 | 462.1 | 489.2 | 510.7 | 454.9 | 493.2 | 23.0 | 8.1 | 4.7 | 474.0 | 512.4 |
| Maximum CSF pressure in CA under BC "A" (Pa) | 565.3 | 640.3 | 701.2 | 665.1 | 570.3 | 580.6 | 703.1 | 557.8 | 623.0 | 61.8 | 21.9 | 9.9 | 571.3 | 674.7 |
| Maximum CSF pressure in CA under BC "B" (Pa) | 569.5 | 597.4 | 636.8 | 583.3 | 492.1 | 529.3 | 598.8 | 485.1 | 561.5 | 54.2 | 19.2 | 9.7 | 516.2 | 606.9 |
| Maximum CSF pressure in CA under BC "C" (Pa) | 481.2 | 502.4 | 508.3 | 494.1 | 453.1 | 481.3 | 503.2 | 449.4 | 484.1 | 22.6 | 8.0 | 4.7 | 465.3 | 503.0 |
| Volume of ventricular system (ml) | 20.1 | 20.9 | 21.8 | 19.89 | 20.1 | 19.6 | 20.9 | 19.5 | 20.3 | 0.8 | 0.3 | 3.9 | 19.7 | 21.0 |
| Volume of brain tissue (ml) | 1275.1 | 1343.2 | 1369.1 | 1338.1 | 1305.5 | 1259.2 | 1210.3 | 1275.1 | 1297.0 | 52.1 | 18.4 | 4.0 | 1253.4 | 1340.5 |
| Volume of SAS (ml) | 100.5 | 109.4 | 111.1 | 105.8 | 101.9 | 113.1 | 103.5 | 101.5 | 105.9 | 4.8 | 1.7 | 4.5 | 101.8 | 109.9 |

BC boundary condition, CSF cerebrospinal fluid, CA cerebral aqueduct, SAS subarachnoid space.

**Table 3 The maximum CSF pressure details and volumes of head substructures of the 11 patients.**

| Hydrocephalus patients Variables | No. 1 | No. 2 | No. 3 | No. 4 | No. 5 | No. 6 | No. 7 | No. 8 | No. 9 | No. 10 | No. 11 | Mean | Standard deviation | Standard error | Coefficient of variation (%) | 95% Confidence interval Lower | Upper |
|---|---|---|---|---|---|---|---|---|---|---|---|---|---|---|---|---|---|
| Maximum CSF pressure in SAS under BC "A" (Pa) | 3730.1 | 3810.5 | 3730.4 | 3994.6 | 3720.2 | 3892.3 | 3540.0 | 3030.7 | 3066.4 | 3170.2 | 3830.3 | 3592.3 | 344.0 | 103.7 | 9.6 | 3361.2 | 3823.4 |
| Maximum CSF pressure in SAS under BC "B" (Pa) | 3145.2 | 3095.3 | 2920.7 | 3189.6 | 2976.5 | 2815.1 | 2898.0 | 2691.2 | 2716.5 | 2943.4 | 3717.7 | 3009.9 | 284.7 | 85.8 | 9.5 | 2818.7 | 3201.2 |
| Maximum CSF pressure in SAS under BC "C" (Pa) | 2559.0 | 2635.0 | 2416.0 | 2659.0 | 2415.0 | 2495.0 | 2400.0 | 2390.0 | 2405.0 | 2670.0 | 2637.0 | 2516.5 | 117.1 | 35.3 | 4.7 | 2437.8 | 2595.1 |
| Experimental value of CSF pressure in SAS (Pa) | 2705.0 | 2795.0 | 2562.0 | 2798.0 | 2543.0 | 2641.0 | 2494.0 | 2524.0 | 2555.0 | 2768.0 | — | 2638.5 | 118.8 | 37.6 | 4.5 | 2638.5 | 2638.5 |
| Maximum CSF pressure in CA under BC "A" (Pa) | 3700.2 | 3803.1 | 3718.9 | 3992.2 | 3713.4 | 3805.3 | 3532.0 | 2992.2 | 3054.1 | 3121.1 | 3825.2 | 3568.9 | 348.7 | 105.1 | 9.8 | 3334.6 | 3803.1 |
| Maximum CSF pressure in CA under BC "B" (Pa) | 3100.3 | 3050.2 | 2818.7 | 3147.6 | 2988.6 | 2830.5 | 2890.0 | 2682.3 | 2670.5 | 2942.4 | 3712.1 | 2984.8 | 287.3 | 86.6 | 9.6 | 2791.8 | 3177.9 |
| Maximum CSF pressure in CA under BC "C" (Pa) | 2538.1 | 2619.5 | 2408.2 | 2645.7 | 2406.5 | 2477.3 | 2386.0 | 2381.2 | 2390.7 | 2650.1 | 2635.3 | 2503.5 | 115.9 | 35.0 | 4.6 | 2425.6 | 2581.4 |
| Volume of ventricular system (ml) | 287.1 | 286.6 | 286.3 | 301.5 | 278.7 | 270.1 | 273.0 | 268.2 | 270.3 | 298.8 | 277.1 | 281.6 | 11.5 | 3.5 | 4.1 | 273.9 | 289.3 |
| Volume of brain tissue (ml) | 1108.2 | 1108.6 | 1092.7 | 1130.6 | 1096.1 | 1018.3 | 1010.0 | 1009.2 | 1009.5 | 1109.1 | 1096.2 | 1071.7 | 48.6 | 14.7 | 4.5 | 1039.0 | 1104.3 |
| Volume of SAS (ml) | 110.1 | 114.0 | 108.1 | 113.1 | 109.3 | 106.7 | 103.5 | 102.0 | 101.0 | 114.8 | 109.8 | 108.4 | 4.7 | 1.4 | 4.4 | 105.2 | 111.6 |

CA cerebral aqueduct, BC boundary condition, CSF cerebrospinal fluid, CA cerebral aqueduct, SAS subarachnoid space.

It should be noted that the *P* values of variance equality in all Student's *t* tests were >0.59.

The results showed that the difference between the experimental and computer-simulated values of the maximum CSF pressure in SAS in patient No. 1–10 was <4.6% under BC "C" (Table 3). However, this difference was more significant than 36.2% under BC "A" and more significant than 14.1% under BC "B" (Table 3). Thus, the data of the maximum CSF pressure in SAS based on BC "C" were closer to the experimental values of the maximum CSF pressure in SAS.

**Changes in the size of head substructures.** Hydrocephalus is caused by an imbalance of production and reabsorption in CSF that involves ventriculomegaly and CSF accumulation in the brain's extraventricular space[42,43]. Therefore, investigating the geometric and volumetric changes of the cranium is of great importance[44]. Furthermore, specialists make their primary decisions based on apparent volume changes in the head substructures diagnosed by visual examination of MRI images. For this reason, the volumes of various head parts in the groups of patients and healthy subjects were compared in this section. It should be noted that these volumes were calculated after 3D modeling of the geometry of the head substructures in CATIA software before analysis. Therefore, these data are independent of the applied BCs "A", "B", and "C". Tables 2 and 3 present the volumes of various head parts in the patients and healthy subjects. The mean values of the ventricular system's volumes, brain tissue, and SAS in the group of patients were 281.6 ± 3.5, 1071.7 ± 14.7, and 108.4 ± 1.4 ml, respectively. The CV of the volumes in both groups was <4.5%. Based on the SD, SE, and CV values of the volumes and the results of ANOVA and Student's *t* tests, the dispersion of the volume data in head substructures of healthy subjects and patients was acceptable.

The mean value of the ventricular system volume in the patients was 13.9 times the corresponding volume in the healthy subjects (Tables 2 and 3). The mean value of the brain tissue volume in the patients was 17.4% less than that in the healthy subjects (Tables 2 and 3). Furthermore, SAS had the least volume change, so that the SAS volume increased <2.4% in the patients (Tables 2 and 3).

**Discussion**
In this research, the inlet/outlet BCs were defined in terms of the CSF pressure and CSF flow rate based on the previous studies and in vivo examinations. However, in general, one can use other definitions for flow rate or pressure and consider them for the inlet/outlet BCs. Therefore, any discussions in this section are purely for the specific inlet/outlet BCs defined in this research.

This part is dedicated to analyzing and comparing the effects of changing the inlet/outlet BCs on CSF flow dynamics. The results showed that the difference between the maximum aqueductal CSF stroke volumes under the three BCs was <0.9%. The assessment of the maximum Reynolds number value of the CSF flow under the three BCs revealed that the CSF flow remained in the laminar phase after the occurrence of hydrocephalus disease and that the difference between the maximum Reynolds numbers calculated under BCs "A", "B", and "C" was <4.4%. The difference between the maximum CSF velocities calculated under BCs "A", "B", and "C" was insignificant (<3.3%) (Fig. 3a–c and Supplementary Data 2). The results also showed that the difference between the maximum values of experimental and biomechanical simulated velocity was acceptable (<4.8%). These results indicated that the inlet/outlet BCs had no significant effect on determining the maximum aqueductal CSF stroke volume, the CSF flow

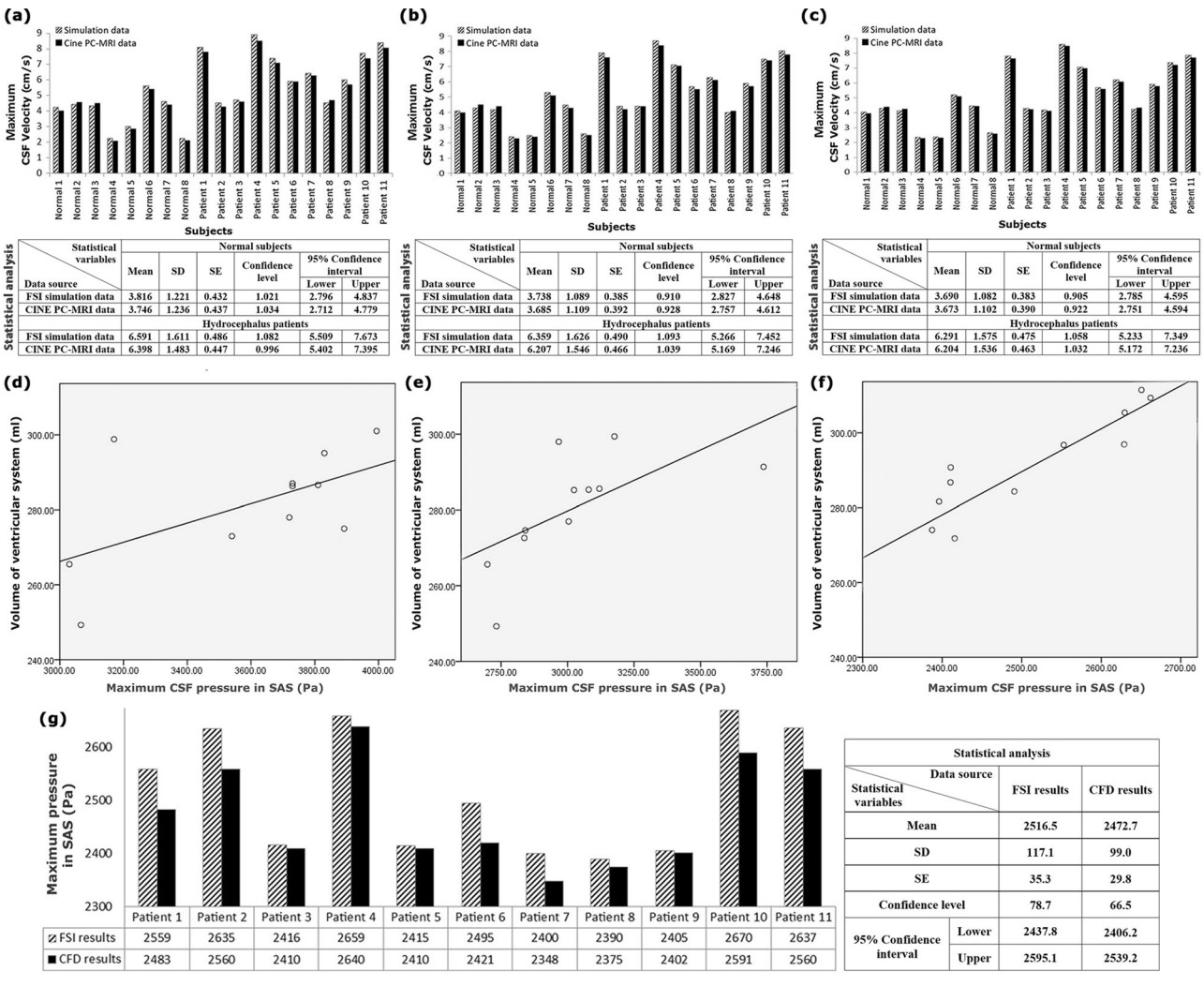

**Fig. 3 Comparisons of the FSI simulation and CINE PC-MRI velocity data, FSI and CFD pressure data, and correlation results. a–c** The comparison between the FSI simulation data and CINE PC-MRI data of the maximum CSF velocity in the CA of all healthy subjects and patients under BCs "A", "B", and "C", respectively. **d–f** The PCC values between the maximum CSF pressure in SAS and the ventricular system volume of patients under BCs "A", "B", and "C", respectively. **g** The comparison between the FSI and CFD data of the maximum CSF pressure in SAS of all the patients under BC "C". It should be noted that there were 8 healthy subjects and 11 hydrocephalus patients in this study. Raw data for **a–c** are included in Supplementary Data file 2. Raw data for **d–g** are included in Tables 2 and 3. SE standard error, SD standard deviation, BC boundary condition, CSF cerebrospinal fluid, CA cerebral aqueduct, SAS subarachnoid space, CFD computational fluid dynamics, FSI fluid–structure interaction, PCC Pearson correlation coefficient.

regime (laminar or turbulent flow), and the maximum CSF velocity.

Comparison of the results of all the parameters calculated under all the three inlet/outlet BCs showed that one of the highest differences between the data of the patients and healthy subjects was related to the maximum CSF pressure (5.1–5.7 times) and the ventricular system volume (13.9 times). The investigation of the CVs of the maximum pressure data and the ventricular system volume also showed a small dispersion in these two parameters. Thus, these two parameters can be considered more effective and accurate for analyzing conditions of hydrocephalus patients. The Pearson correlation coefficient (PCC) was used in this study to assess the correlation between the maximum CSF pressure in SAS and the ventricular system volume (two effective and accurate indices) under the three inlet/outlet BCs. As observed in Fig. 3d–f, the values of the PCC of patients under the inlet/outlet BCs "A", "B", and "C" were 0.47 ($P < 0.0046$), 0.66 ($P < 0.0041$), and 0.81 ($P < 0.0011$), respectively. PCC in BCs "A" and "B" was, respectively, 42.0% and 18.5% lower than that in BC "C".

Moreover, the weakest correlation between the maximum CSF pressure in SAS and the ventricular system volume occurred under BC "A". Accordingly, in constant pressure changes, the biomechanical model was less sensitive to the ventricular system volume changes under BC "A" compared to the other inlet/outlet BCs.

The ventricular system volume, as mentioned above, was not related to our inlet/outlet BCs and was measured in CATIA before analysis. Therefore, only the difference between maximum CSF pressure under the three BCs led to a difference in PCCs. There was also a significant nonconformity between the maximum CSF pressure in SAS, which is obtained experimentally from ICP monitoring, and the FSI simulation results of CSF pressure in SAS under BCs "A" and "B". These nonconformities in BCs "A" and "B" were 36.2% and 14.1%, respectively (Table 3). The biomechanical simulation results of CSF pressure in SAS under BC "C", however, were close to the maximum CSF pressure in SAS, which is measured experimentally from ICP monitoring, with an error of <4.6% (Table 3). The data dispersion and CV of

the maximum CSF pressure in SAS under BCs "A" and "B" was also more than twice the data dispersion of the maximum CSF pressure in SAS under BC "C" (Table 3). These examinations revealed that the inlet/outlet BCs "A" and "B" caused a significant error in calculating the maximum CSF pressure compared to the experimental data. The correlation of CSF pressure in SAS with the ventricular system volume in BC "C" was also at least 18.5% higher than that in BCs "A" and "B" (Fig. 3d–f). Thus, the inlet/outlet BC "C" could be more accurate for obtaining the best correlation between the maximum CSF pressure in SAS and the ventricular system volume and evaluating intracranial compliance (ICC) during the treatment process of hydrocephalus patients. The reason is that the manner of changes in the pressure–volume diagram ($\Delta V/\Delta P$) slope during hydrocephalus patients' treatment process indicates changes in these patients' ICC during the healing process[45,46]. It is noteworthy that ICC is a highly valuable index for assessing hydrocephalus patients[32,47].

Knowing the exact amount of ICP can help understand the biomechanical properties of brain tissue[48]. As mentioned before, the ventricular system volume and CSF pressure in SAS can reveal ICC, a valuable index for analyzing hydrocephalus patients' conditions during the treatment process. However, on the one hand, experimental methods for measuring CSF pressure like ICP monitoring or lumbar puncture, which are also invasive or minimally invasive techniques, were not recommended for measuring CSF pressure in all types of hydrocephalus[5,46]. On the other hand, it is not always possible to measure CSF pressure at all locations, such as CA or optic nerve sheath, using these experimental techniques[5]. The computer simulation-based analysis in this study can calculate the velocity, flow rate, stroke volume, and CSF pressure at all cross sections of the head noninvasively.

The impact of BCs is not limited to the inlet/outlet BCs, and the interface between fluid and solid models is also a useful parameter of BCs in a variety of solution methods like CFD and FSI[49]. Therefore, in this section, the effect of different solution methods with various fluid–solid interfaces was examined and compared on the analysis of CSF dynamics in internal and external hydrocephalus.

The effect of CSF flow was ignored in previous FEM studies[50–52]. The FEM solution can only be used for biomechanical simulation of solid models[53] like a skull. In these studies, the interaction between the CSF interfaces and the internal and external brain tissue walls was not considered. Therefore, the results calculated from separate brain tissue analyses without considering the CSF effect are not comprehensive for evaluating hydrocephalus patients. In FEM studies, the CSF effect is often replaced with a constant distributed compressive load[12]. However, the fluid flow is pulsatile and highly complex, and thus, replacing it with a distributed compressive load cannot fulfill the requirements for flow analysis. Therefore, the FEM solution cannot be used for CSF flow analysis in hydrocephalus patients. Thus, this solution method was not examined in this study.

The results showed that the smallest volume change in the head occurred in SAS. The SAS volume change was <2.3% and could be neglected. The main difference in using the CFD and FSI methods to solve engineering problems is whether the fluid–solid boundary is deformable[49]. The question arising now is whether it is necessary to use the FSI method, while the difference between these two methods is merely based on the deformability of BC at the interface between fluid and solid, and the deformation of these boundaries is negligible in SAS.

For this reason, only the CSF flow in SAS of all the patients was analyzed in this step using the CFD method. The results above revealed BC "C" to be more accurate for evaluating CSF, and hence, the CSF pressure values calculated in SAS using the CFD

method were compared with those calculated using the FSI method. The no-slip BCs governing Eq. (6) were assumed for all SAS's inner and outer surfaces. The inlet and outlet flow-rate functions were completely similar to those in BC "C", and the outlets' locations were defined as those in BC "C". However, the CSF inlet location was defined after the foramina of Luschka and Magendie (the inlet of the SAS area). The problem was then analyzed using the Navier–Stokes equation (Eq. (1)). The results showed that the maximum CSF pressure calculated in SAS using CFD was in agreement with that calculated using FSI with a maximum difference of 3.0% (Fig. 3g), although the solving time in CFD was reduced to 7.1 times. It is noteworthy that the phase difference was minimal between the CSF pressure diagrams in SAS calculated using these two methods in all the patients; the maximum amount of this difference was 0.4%. Therefore, CFD is a sufficiently accurate CSF flow analysis method in SAS since the fluid–solid interface in SAS—neglecting the small volume change—has no deformation, and there is no need to use the FSI method.

Thus, the CFD method can be used to calculate CSF pressure in SAS for external hydrocephalus, such as Chiari malformation and pseudotumor cerebri, which occurs due to a cause outside the ventricular system. In contrast, in internal hydrocephalus diseases, such as aqueductal stenosis, mesencephalic tumor, and aqueductal web, it is necessary to solve the FSI problem to calculate CSF pressure in SAS due to significant volume changes of the ventricular system (the ventricular system volume increased 13.9 times after the disease occurrence) and the significant deformation of the fluid–solid boundary.

As one of the most significant parameters in the evaluation of hydrocephalus patients, measuring the maximum CSF pressure is not possible using current experimental noninvasive methods. The present noninvasive biomechanical simulations are useful for accurately measuring the CSF hydrodynamic parameters, such as CSF pressure changes, using the suggested BC and solving method, which help diagnose[49] and treat[33] hydrocephalus patients.

Brain tissue was considered a homogeneous material in the form of a single-phase model in this study, and the results showed that considering this model for brain tissue was accurate enough to analyze the patients' conditions. Dutta-Roy et al. showed no significant difference in the ventricular system volume calculation in normal pressure hydrocephalus patients using a biphasic and single-phase model[54]. On the other hand, in Lefever et al.'s study, a noncommunicating hydrocephalus patient's brain tissue was simulated with a biphasic model[55]. Therefore, it is suggested that future studies evaluate and compare the impact of single-phase or biphasic brain tissue models[49], and also, the effect of fusing the cerebellum with the cerebrum under the BCs examined in this study.

Classical studies have shown that the brain acts like a sponge[56,57]. This notion led to the use of the porous media theory for brain biomechanics[3,58]. Later, concepts beyond the passive sponge concept were proposed for brain tissue[3,58]. Many studies, such as the present one, used Darcy's law for the biomechanical description of brain tissue, based on the study by Cheng et al.[14]. However, Tavner et al. indicated that Darcy's law might be inadequate for describing the brain tissue constitutive behavior[59]. There exist many contradictions and conflicts in using hyperelastic models[49,59,60] or models based on Darcy's law[49,59,61,62] for modeling the brain during the biomechanical simulations of the hydrocephalus. Hence, it is suggested to develop advanced theoretical models for a more accurate description of the brain's constitutive behavior during the biomechanical simulation of hydrocephalus patients in future studies.

Park et al. revealed that the cardiac pulsation absorbance went down significantly after chronic hydrocephalus in animal subjects[63]. They also found that a significant part of this problem with hydrocephalus was the disrupted input flow to microvessels in the brain. Regarding the approach and goals of the present simulation study, this phenomenon could not make a proper assessment. Hence, future studies are suggested to consider this effect in the inlet BC and assess changes in cardiac pulsation absorbance in the simulation results using a microscopic approach to utilize this parameter as complementary metrics in the evaluation of hydrocephalus patients. Future studies are also recommended to evaluate the relationship between BCs and symptom severities in larger sample sizes.

The 3D biomechanical simulation of the ventricular system, SAS, and brain tissue was performed in 11 hydrocephalus patients and eight healthy subjects. Moreover, the dynamic interaction of the CSF flow was investigated under three different inlet/outlet BCs. The results of examining the parameters of the aqueductal stroke volume, the Reynolds number, CSF velocity, and pressure and evaluating the volumes of the head substructures showed that the maximum CSF pressure and the ventricular system volume were the more effective and accurate indices for evaluating conditions of hydrocephalus patients. The results also showed that among the three BCs examined in this study, BC "C", considering the pulsatile CSF flow-rate diagram as the inlet and outlet, was more accurate for analyzing the maximum CSF pressure in SAS. The volume changes in SAS after the disease occurrence were small, and the analysis of the CSF flow in SAS showed that the CSF pressure values calculated in SAS using the CFD and FSI solution methods agreed with each other and had only an insignificant difference. The results also showed the necessity of using the FSI method only when the disease's cause was inside the ventricular system.

## Methods

Previous studies have shown that gender and age affect the biomechanical properties of brain tissue in hydrocephalus patients and healthy subjects[64–66], and this has a significant effect on biomechanical simulation results. Consequently, there were many limitations in selecting healthy subjects and patients in the present study. Accordingly, 11 patients and eight healthy subjects were recruited from 42 hydrocephalus patients and 23 healthy subjects. The patients (six males and five females) and healthy subjects (four males and four females) had an age range of 53–74 and 48–69 years, respectively. Moreover, the body mass index range of the patients and healthy subjects was 23.8–29.4 and 25.0–30.2, respectively. CINE PC-MRI was performed on the head in all the healthy subjects and patients (Fig. 4a–c).

All procedures were under the Shohada Tajrish Hospital's ethical standards and the 1964 Helsinki declaration and its later amendments or comparable ethical standards. The Ethics Committee of the Functional Neurosurgery Center at the Shohada Tajrish Hospital approved the study design, procedures, and protocols (the ethics number 18/54-9). Furthermore, this paper did not include any studies on animals. All the patients and healthy subjects provided verbal informed consent before undergoing any study-specific procedures.

A velocity-encoding value (VENC) of 100 cm/s was chosen to measure the CSF flow. Further parameters used in the measurement included repetition time = 18 ms, flip angle = 23°, echo time = 8.3 ms, field of view = 23 cm, slice thickness = 3 mm, and matrix size = 256 × 198. The pixel velocity in CSF areas was corrected by subtracting the average velocity of solid brain tissue in a nearly 29 × 29-mm² area surrounding the pixel. More details on the imaging protocol are explained by Kapsalaki et al.[67]. Scanning was performed using a 3 Tesla MRI system (Magnetom Trio, Siemens, Erlangen, Germany), with an acquisition time of 45 min.

DICOM files obtained from MRI of each healthy subject and patient were transferred to Mimics software v13.1 to prepare the points cloud that the points in the cloud were voxel centers. Delineation of boundaries between CSF spaces and brain tissue was also performed using Mimics software. The point clouds of the head substructures (SAS, brain tissue, and ventricular system) were produced for each healthy subject and patient and transferred to CATIA v5.R21 for 3D geometrical modeling. After creating 3D geometrical models of the head of the healthy subjects and patients separately (Fig. 4d), the models were transferred to ADINA 8.3 (Adina R&D Inc., Watertown, MA, USA) for meshing (Fig. 4e) and analysis.

CINE PC-MRI was used as a measurement tool for the in vivo investigation of the CSF flow inside CNS[67]. Methods based on medical imaging have had many improvements recently; however, they cannot be applied for CSF pressure measurement. CSF pressure is measured with invasive methods such as ICP

monitoring. However, these invasive methods cannot measure CSF pressure at all CSF circulation locations in the head[5,6]. Thus, it is required to use computerized simulation models for the quantitative assessment of hydrocephalus. In the present study, the CSF velocity data measured at the CA of all the healthy subjects and patients with the help of CINE PC-MRI were compared with the similar velocities calculated by biomechanical simulations, and the blood-flow rate data measured at the basilar artery of all the healthy subjects and patients with the help of CINE PC-MRI were used for inlet/outlet BCs.

**Numerical simulations.** Simulations were performed using a fully coupled FSI (two-way FSI with strong coupling) model, in which arbitrary Lagrangian–Eulerian (ALE) equations were used for the simultaneous analysis of fluid (CSF) and structural (brain tissue) models. In an FSI problem, the computational grid of the fluid model deforms. Hence, in the ALE formulation, velocity relative to mesh velocity is used in the momentum equation's convective term. It means that velocity at the faces of a moving and deforming control volume is the relative velocity concerning the faces of the control volume (or the computational grid)[68]. An iterative solution was employed to solve the coupling of fluid and structure models[68]. Variables in this solution were fully coupled and defined at the element center. The velocity–pressure coupling was handled iteratively. The equations of the fluid and structure domains were solved individually using the counterpart equation's latest results in the coupled system. The iteration process continued until the acceptable convergence of the results was achieved. It should be noted that the CSF pressure values obtained with ADINA software were calculated in Gauss points[69].

Continuity governs the fluid flow, and Eq. (1) is the Navier–Stokes equation for the CSF flow [70–75]:

$$\rho_F \frac{\partial \boldsymbol{u}_F}{\partial t} + \rho_F((\boldsymbol{u}_F - \boldsymbol{W}).\nabla)\boldsymbol{u}_F = -\nabla p + \mu \nabla^2 \boldsymbol{u}_F + \boldsymbol{f}_F^B \qquad (1)$$

where the terms $\boldsymbol{W}$, P, $f_F^B$, and $\mu$ are the moving mesh velocity vector, CSF pressure, body force per unit volume, and CSF dynamic viscosity, respectively, $u_F$ is the CSF velocity vector, $\rho_F$ is the CSF density, and $(u_F$-$\boldsymbol{W})$ in ALE is the relative fluid velocity concerning the moving coordinate velocity. Equation (2) governs the solid domain[68]

$$\nabla.\sigma_S + \boldsymbol{f}_F^B = \rho_S \ddot{\boldsymbol{u}}_S \qquad (2)$$

where $\rho_S$ and $\ddot{u}_S$ are the solid density and the local acceleration of the solid part, respectively.

The head geometrical model created for analysis was composed of three substructures: SAS, brain tissue, and the ventricular system. According to previous studies, the skull and dura mater's role was negligible in the hydrocephalus biomechanical simulation[27–33]. Therefore, they were ignored in this study's biomechanical simulation process.

It is noteworthy that boundaries between brain tissue, the ventricular system, and the skull are highly complex and significant[76]. The interface between the ventricular system's outer surfaces and brain tissue's inner surfaces was defined as the FSI boundary. It has been demonstrated in Gholampour's study that the deformation of SAS's outer surfaces in all directions is negligible and that defining the FSI boundary for SAS's outer surfaces has almost no considerable effect on the results[33]. Therefore, in the present study, the interface between brain tissue's outer surfaces and SAS's inner surfaces was defined as the FSI boundary, and no-slip BC was applied to SAS's outer surfaces. The following equations formulate the coupling of the domains, which were described in Eqs. (1) and (2), through displacement compatibility and traction equilibrium[70,77,78]

$$\boldsymbol{d}_S = \boldsymbol{d}_F \quad (x, y, z)\epsilon\Gamma_{\text{wall}}^F \cap \Gamma_{\text{wall}}^S \qquad (3)$$

$$\sigma_S.\boldsymbol{n} = \sigma_F.\boldsymbol{n}(x, y, z)\epsilon\Gamma_{\text{wall}}^F \cap \Gamma_{\text{wall}}^S \qquad (4)$$

$$\boldsymbol{u}_S = \boldsymbol{u}_F(x, y, z)\epsilon\Gamma_{\text{wall}}^F \cap \Gamma_{\text{wall}}^S \qquad (5)$$

where $\boldsymbol{d}_S$, $\boldsymbol{d}_F$, $\sigma_S$, and $\sigma_F$ are the brain tissue displacement, CSF flow displacement, brain tissue stress tensor, and CSF stress tensor along with the fluid-structure interface on FSI boundary, respectively, $\Gamma_{\text{wall}}^S$ and $\Gamma_{\text{wall}}^F$ are the boundaries of the solid and fluid domains, and n is the normal direction of the fluid-structure interface. Equation (3) confirms the displacement compatibility between CSF and brain tissue along with the fluid-structure interface. Equation (4) is called dynamic BC or the traction equilibrium equation and formulates the equilibrium condition between stresses acting in normal direction on both domain boundaries $\Gamma_{\text{wall}}^S$ and $\Gamma_{\text{wall}}^F$. Equation (5), fluid velocity BC, shows that the velocity of solid walls is equal to the fluid velocity at the interface. It is noteworthy that the BCs mentioned above govern the interface between brain tissue's and CSF's inner and outer surfaces, i.e., the FSI boundary. According to Eq. (6)[79–81], however, no-slip BCs govern nondeformable interfaces (outer surfaces of SAS).

$$\boldsymbol{u}_F = 0 \qquad (6)$$

The density and dynamic viscosity of CSF were assumed as those of water at 37 °C, i.e., 998.2 kg/m³ and 0.001 kg m⁻¹ s⁻¹, respectively[15,27–30]. Furthermore, CSF was defined as a Newtonian fluid.

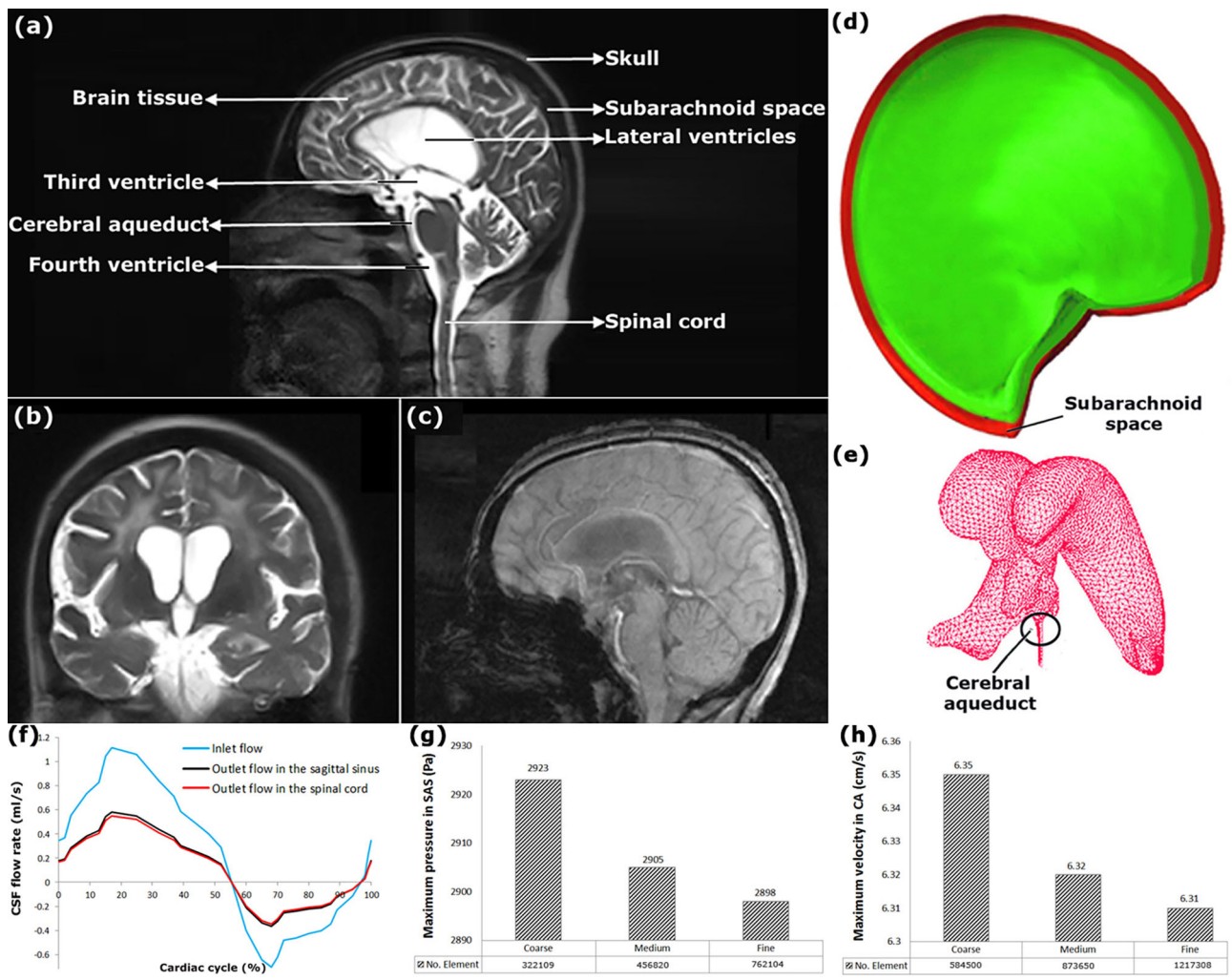

**Fig. 4 3D geometrical models, boundary conditions, and mesh convergence study data. a** shows the locations of the head substructures. **b**, **c** show the head MRI images for patient No. 7. **d** shows the 3D geometrical model of SAS for patient No. 7. **e** shows mesh modeling of the ventricular system. **f** shows the inlet and outlet flow-rate diagrams in BC "C". **g**, **h** show mesh convergence study for the maximum CSF pressure in SAS and the maximum CSF velocity in CA under the BC "B" of the patient No. 7, respectively. BC boundary condition, CSF cerebrospinal fluid, CA cerebral aqueduct, SAS subarachnoid space.

In previous biomechanical simulations, various constitutive models were considered for brain tissue. However, it has been demonstrated that brain tissue shows time-dependent stress–strain behavior[82]. Linninger et al. and Sweetman et al. assumed poroelastic parenchyma in their computer simulations[27–29,31]. In other studies, viscoelastic and hyperviscoelastic models have been used for brain tissue simulation[30,32,83,84]. Among numerous assumptions used in previous studies, the assumption of poro-viscoelastic material for brain tissue in biomechanical simulation by Cheng et al. had a better agreement with previous experimental studies[14]. Hence, the same assumption was made for brain tissue in the present study. The solid model equation, the law of Darcy for the flow of a fluid through a porous medium, and the stress field's equilibrium conditions were used to derive general equations according to Cheng et al.[14]. The relaxation modulus was another parameter of the poro-viscoelastic model[14]. The Prony series was used to express the constitutive relation of the viscoelastic solid phase[14]. The equation of the time-dependent shear modulus of relaxation is as follows:

$$\mathrm{Gr}(t) = \mathrm{G}_0\left(1 - \sum_{k=1}^{N} g_k^{-p}\left(1 - e^{-\left(\frac{t}{\tau_k}\right)}\right)\right) \quad (7)$$

where $g_k^p$ and $\tau_k$ indicate the Prony series's input parameters dominating the relaxation response and $\mathrm{G}_o$ represents the instantaneous shear modulus.

Taylor et al. suggested the value of 584.4 Pa for the elastic modulus of hydrocephalus patients' brain tissue[12]. Other studies have used values of 350.0 and 420.6 Pa for elastic modulus[14,85,86]. There was a good agreement between the subsequent experimental studies and the study of Taylor et al. performed on hydrocephalus patients. Therefore, the elastic modulus of brain tissue used the value of 584.4 Pa in this study. There is also a large variation in the brain tissue

Poisson's ratio used in different studies. Some studies have considered the value of 0.45–0.5 for the Poisson's ratio[28,29,65]. In this study, however, the Poisson's ratio was assumed 0.35 based on Dutta-Roy et al.'s and Cheng et al.'s studies due to the better agreement with the subsequent experimental studies[14,54,86]. Further, the void ratio and permeability values were defined as 0.2 and $4.08 \times 10^{-12}$ M⁴/N s, respectively[14,54]. The input parameters of the Prony series were also selected according to studies by Dutta-Roy et al. and Cheng et al.[14,54]: $g_k^p = 0.285$, $\tau_1 = 3.1$ (s), $\tau_2 = 27$ (s), $\tau_3 = 410$ (s).

**Inlet/outlet boundary conditions of models**. As mentioned before, pressure and velocity or the CSF flow rate was defined as inlet/outlet BCs in previous studies. Zero pressure was considered as the outlet BC in many previous CFD simulations[16,17,22,23,26]. According to the study by Marsden et al., this assumption is not sufficient for solving the problem using the FSI method[10]. The conclusion of the previous BCs is summarized in Table 1. Accordingly, the following three inlet/outlet BCs were selected for the FSI simulation of all the healthy subjects and patients in this study:

BC "A": A sinusoidal pressure function was used at the inlet and outlet BCs to mimic CSF's normal oscillation. In BC "A", the inlet pressure function was defined as $80 + 160 \sin(2\pi t)$ Pa, according to the study by Jacobson et al.[17]. Baseline pressure in the sagittal sinus as the outlet pressure function was 500 and 2700 Pa in the healthy subjects and patients, respectively[28,30,31].

BC "B": The pulsatile blood flow profile was always considered in the literature as the inlet BC, and the pressure value was imposed at the outlet BC[27–32]. In this study, the CSF pulsatile flow rate profile was used as the inlet BC in BC "B", and the outlet BC was defined similarly to that in BC "A". The inlet BC was calculated with the superposition of a constant value of 0.35 ml/min[1,87] and the normalized

pulsatile profile of the blood flow rate in the basilar artery, which was measured with the CINE PC-MRI for all the healthy subjects and hydrocephalus patients.

BC "C": CSF is produced mainly in the choroid plexus, and is absorbed in various locations, i.e., with arachnoid granulations[49,88,89], at extracranial lymphatic pathways[49,90], and in additional intraparenchymal routes[49]. After CSF is produced and passed through the lateral, 3rd and 4th ventricles, it flows through a series of apertures into SAS, which surrounds brain tissue between the pia mater and the arachnoid mater[1]. In SAS, part of the fluid enters the sagittal sinus[91], and the rest drains through the central canal into the spinal cord[92]. Of course, the drainage of CSF also occurs through other parts like small granulation tissues of arachnoids. As in previous computer simulations of hydrocephalus, the primary inlet flow was considered in lateral ventricles, whereas the significant outlet flow was considered in the sagittal sinus and the spinal cord[93–95]. These locations were merely defined in BC "C" as locations to apply the inlet and outlet BCs. In BC "C", the inlet and outlet flow diagrams were pulsatile CSF flow-rate diagrams (Fig. 4f), and the inlet BC was similar to that in BC "B".

Each of the inlet/outlet flow-rate diagrams was calculated with the superposition of two diagrams in MATLAB. One diagram was the amplitude diagram, which was a constant diagram with the value of 0.35 ml/min for the inlet flow based on physiological data[1,87] and the values of 0.18 ml/min and 0.17 ml/min for the outlet flow in the sagittal sinus and the spinal cord, respectively[96]. The other diagram was obtained by normalizing the blood-flow rate function in the basilar artery measured from the CINE PC-MRI of all the healthy subjects and patients.

**Mesh convergence study**. Concerning element dimension, angle, and warpage, regular meshing is used in this study. Regular meshing can generate meshes with similar sizes and uniform distribution[97]. An implicit Euler scheme was used to refine the grids successively with a time step of 0.01 through transient analysis. The mesh convergence study was conducted by analyzing the mesh density effect on the maximum CSF pressure in SAS and the maximum CSF velocity in CA, calculated using ADINA. The maximum difference between the medium and fine meshes in both CSF velocity and pressure data for all the healthy subjects and patients was about 0.31%. Hence, a convergence of the results was achieved regarding mesh density (Fig. 4g, h and Table 4).

It should be noted that a four-node tetrahedral element was used to mesh the SAS and ventricular system of the healthy subjects and patients under all BCs. Moreover, an eight-node hexahedral element was used to mesh the brain of the healthy subjects and patients. Furthermore, the integration scheme used in the study was fully integrated. Table 4 shows the parameters' values and the number of meshes in all the patients in three computational grids under BC "C". It is worth noting that the meshes had a higher density at CA in the healthy subjects and patients.

**Statistics and reproducibility**. Since the patients' data before contracting hydrocephalus were not available, a group of healthy subjects with similar age and gender and of almost similar height and weight to the patients was recruited in the study to assess the deviation of patients' conditions from normal conditions. A statistical dataset such as aqueductal CSF stroke volumes, the Reynolds number, velocities, pressures, and volumes was required to be assessed for the healthy subjects and patients. Thus, statistical variables such as mean, SD, and SE were calculated for each dataset in healthy subjects and patients under each BC ("A", "B", and "C") using IBM SPSS Version 20 software. Generally, the mere investigation of mean, SD, and SE is not sufficient to compare some statistical datasets, and the comparison of data dispersion is also necessary. CV as the standard deviation ratio to the data mean value is a useful statistical index to compare the data dispersion between different data series[98]. Accordingly, CV for each dataset in the groups of healthy subjects and patients was calculated separately for each BC, in addition to the aforementioned statistical variables.

The results of the Shapiro–Wilk test showed that the datasets had a normal distribution. Parametric ANOVA multiple comparison was used[99] to compare the CSF pressure, the aqueductal CSF stroke volume, the Reynolds number, the CSF velocity, and the volume between healthy subjects and patients under the three BCs "A", "B", and "C". The homogeneity of the variance test showed that all the variances were equal. Hence, Tukey's post hoc test was used for pairwise comparison after ANOVA when comparing the data under the three BCs. Moreover, the normally distributed data led to using Student's $t$ test after ANOVA to compare both groups of CSF velocities obtained from computer simulation and CINE PC-MRI, and two groups of CSF pressures obtained from computer simulation and ICP monitoring for assessing data validation. Student's $t$ test with equal variance was also used to compare the CSF pressure results of CFD and FSI simulations. The test statistics for ANOVA and Student's $t$ test were T and F, respectively.

The PCC, a number between −1 and +1, is an index for evaluating the correlation between two phenomena[100]. Hence, after ensuring the normal distribution of the data, PCC was used to assess the correlation between the maximum CSF pressure and the ventricular system volume under the three BCs. The data were described as mean ± SE, and the $P$ value of 0.05 was considered statistically significant.

**Table 4 The maximum CSF velocities and pressures of all patients in three computational grids, and number of fine-mesh under BC "C".**

| Hydrocephalus patients | | No. 1 | No. 2 | No. 3 | No. 4 | No. 5 | No. 6 | No. 7 | No. 8 | No. 9 | No. 10 | No. 11 |
|---|---|---|---|---|---|---|---|---|---|---|---|---|
| Maximum CSF pressure in SAS (Pa) | Coarse | 2594 | 2669 | 2450 | 2693 | 2457 | 2538 | 2434 | 2432 | 2441 | 2711 | 2670 |
| | Medium | 2567 | 2643 | 2423 | 2667 | 2421 | 2503 | 2407 | 2397 | 2413 | 2678 | 2645 |
| | Fine | 2559 | 2635 | 2416 | 2659 | 2415 | 2495 | 2400 | 2390 | 2405 | 2670 | 2637 |
| Maximum CSF pressure in CA (Pa) | Coarse | 2571 | 2658 | 2443 | 2681 | 2439 | 2511 | 2420 | 2418 | 2422 | 2687 | 2670 |
| | Medium | 2543 | 2627 | 2411 | 2653 | 2413 | 2484 | 2391 | 2389 | 2397 | 2657 | 2642 |
| | Fine | 2538 | 2619 | 2408 | 2645 | 2406 | 2477 | 2386 | 2381 | 2390 | 2650 | 2635 |
| Maximum CSF velocity in CA (cm/s) | Coarse | 7.9 | 4.38 | 4.27 | 8.7 | 7.19 | 5.76 | 6.28 | 5.3 | 5.99 | 7.4 | 7.91 |
| | Medium | 7.83 | 4.32 | 4.21 | 8.63 | 7.1 | 5.69 | 6.22 | 4.26 | 5.92 | 7.37 | 7.87 |
| | Fine | 7.8 | 4.3 | 4.2 | 8.6 | 7.07 | 5.68 | 6.2 | 4.25 | 5.9 | 7.35 | 7.85 |
| Number of fine-mesh | Ventricles and SAS | 1,280,569 | 1,285,432 | 1,265,957 | 1,308,952 | 1,248,592 | 1,221,508 | 1,221,508 | 1,178,956 | 1,185,963 | 1,315,024 | 1,242,859 |
| | Brain | 778,095 | 779,958 | 778,098 | 788,954 | 780,024 | 765,892 | 761,206 | 758,326 | 759,842 | 785,651 | 779,862 |

BC boundary condition, CSF cerebrospinal fluid, CA cerebral aqueduct, SAS subarachnoid space.

**Reporting summary**. Further information on research design is available in the Nature Research Reporting Summary linked to this article.

## Data availability

The MRI files of subjects contain some identifying information of patients and normal subjects, and cannot be made publicly available. All relevant data are available from the corresponding author upon request. Raw data for Fig. 1a and Fig. 3a–c are included in Supplementary Data files 1 and 2, and raw data for Fig. 3d–g are also included in Tables 2 and 3.

## Code availability

All settings of the software and packages, which are used for the analysis, are publicly available and fully described in "Methods".

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

## Acknowledgements

No funding was solicited for this research. The authors are very grateful to the neurosurgeon team at the hospital Shohada Tajrish in Tehran, for the cooperation in this study. The authors also would like to thank Mansore Gholampour for her advice about statistical analysis.

## Author contributions

S.G. contributed to the conceptualization, design of the study, data curation, formal analysis, investigation, methodology, project administration, resources, software, and validation. N.F. contributed to the design of study, data curation, methodology, and writing the paper. All authors approved the final paper.

## Competing interests

The authors declare no competing interests.
