## [Peer Review File · Communications Biology]

Reviewers' comments:

Reviewer #1 (Remarks to the Author):

This manuscript describes the computational studies on flows of cerebrospinal fluid (CSF) in eight control and eleven individuals with hydrocephalus using finite element analysis software, "ADINA" and CAD software "CATIA", which allow to calculate velocity and pressure of the fluid (CSF) in the cerebral ventricles, aqueduct, and subarachnoid space (SAS). The authors are able to simulate geometries of the CSF flow pathways in the brain. However, authors are not familiar with how to compare or represent human dataset in an expected way, particularly when the disease is involved. I have two major concerns and several minor comments.

Major problems #1.

It seems that there is no validation to show the accuracy of the solution. Only in Table 1, I found a comparison with Experimental values without any reference. In page 19, the authors talked about inlet/outlet BCs and provided few references for that. The Boundary condition strongly depends on the physics of the problem and I doubt if authors can demonstrate exact solutions or experimental results that may be comparable to what you have analyzed using FEA software.

Major problems #2.

Table 1 is utterly wrong. At present form, you provide a twisted impression that there are 11 experimental subjects and 1 control (healthy/normal). According to your abstract, there are 8 control subjects. Therefore, your Table 1 should address 1) either all individuals of N=19 or 2) mean, standard deviation per group. For example,

Suggested new Table 1.

Case number: Control (n=8) Hydrocephalus (n=11)

Max CSF pressure in SAS 650+/-2 3500+/-10
(mean+/-s.e.)

Max CSF pressure in CA 560+/-2 3100+/-11
(mean+/-s.e.)

.....

Minor comments or suggestions:

pp. 8 Line 22 "...Hydrocephalus occurs..."

(suggestion) "...Hydrocephalus results from imbalance of production and reabsorption in CSF, that involves ventriculomegaly and CSF accumulation in the extra-ventricular space of the brain. Therefore, investigating the geometric and volumetric change of the cranium is of great importance.

pp. 21 Statistics

You should rewrite. Describe the method. What stat method did you use? You basically compare two groups in your study. Thus, you should specify if you were using "parametric test such as Student t test or independent t test" or "non-parametric test such as Mann-Whitney U test". Here is an example of how to write a statistical method that you are expected to use:

"...Statistical analysis

Statistical analyses were performed using SigmaPlot (version 11, Systat Software Inc.) and Prism (GraphPad version 7). Normality of data distribution was tested using the F-test for unequal variance. Normally distributed data were analyzed using Student's t-test and Tukey's post-hoc test for pair-wise comparison after ANOVA when comparing two or three groups respectively. Non-normally distributed data were analyzed using the nonparametric Mann-Whitney test and Kruskal-Wallis with Dunnett's post hoc tests when comparing two and three data groups, respectively. Data were expressed as average \pm standard error of the mean (S.E.) and were considered significant at the $p \leq 0.05$ level...." (source: PMID: 30705305)

pp. 33 Fig. 1 ...Panels (e) and (f)...
(...Panels (f) and (g)...

Thank you.

Reviewer #2 (Remarks to the Author):

I think these three boundary condition simulations (A, B, and C) and the cine-flow data collected from 11 hydrocephalus patients and 8 healthy controls could make a useful contribution--and they appear to be novel. However, this paper is probably too long, it needs to add some sort of statistical analysis (e.g., do the boundary conditions fall in the 95% confidence interval of the cine-flow data?), and it would be helpful to readers if the authors framed better in the Discussion section or Conclusion how these results extend our knowledge concerning hydrocephalus.

This paper emphasized CSF pressure in the cerebral aqueduct (CA) and the subarachnoid space (SAS) in terms of characterizing hydrocephalus. However, Park et al. (2010, Journal of Neurosurgery: Pediatrics) used an animal (dog) model in which pre- and post-fourth-ventricle obstruction was used to assess the cardiac pulsation absorbance (CPA) index. They found that the CPA index went down significantly after chronic hydrocephalus was induced in these animals. This suggested that a major part of the problem with hydrocephalus is that the input flow to micro-vessels in the brain is disrupted. How would the present model incorporate the vascular changes in hydrocephalus?

Finally, did the boundary conditions (A, B, or C) or cine-flow data correlate with symptom severity in hydrocephalus patients?

Reviewer #3 (Remarks to the Author):

- 1- The abstract is quite long and densely written. I propose to shorten it and focus on the main message, e.g. FSI model advantages. I would pick up boundary conditions in the discussion, but shorten its significance in the abstract, because the write up is otherwise too technical.
- 2- Intracranial Pressure (ICP) and CSF pressure — it would improve the clarity of the study if the authors could elaborate more on the differences between the ICP and CSF pressure. How the ICP was obtained (computed) in the models?
- 3-The description of the methods of computer simulation is not comprehensive. The information about mesh information is incomplete and the authors should add more details about them.
- 4- Why do you neglect the skull bone in your 3D geometric model. What will be the effect of this on your results?
- 5- The FSI method which used in this study was one-way or two-way? Do you think the other method which you don't use in your study, will have a significant impact on your results? Why?
- 6- Do you re-calculate all models of all samples using CFD simulation as similar to FSI? Clarify this in the main text. Because it is not clear that Fig. 3g is the result of which comparison. Is it the result of the comparison of all samples before and after treatment process?
- 7- Add the names of some types of hydrocephalus which CFD method can be a better method for simulating of them. Also, for FSI method. Clearly add the samples for internal and external hydrocephalus in your discussion section.

Dear Reviewers

We appreciate you and your valuable time and care for reviewing our manuscript. Your point of view was very helpful in improving the quality of this article and clarifying the vague points of the manuscript. Therefore, we have revised the manuscript with respect to all your valuable guides and comments. It should be noted that the manuscript is proofread for English editing by a reputable institute, and the certificate is attached. Furthermore, the texts modified or added to the manuscript have been shown with yellow highlighted.

With best regards

Seifollah Gholampour
Corresponding author

Reviewer #1:

This manuscript describes the computational studies on flows of cerebrospinal fluid (CSF) in eight control and eleven individuals with hydrocephalus using finite element analysis software, “ADINA” and CAD software “CATIA”, which allow to calculate velocity and pressure of the fluid (CSF) in the cerebral ventricles, aqueduct, and subarachnoid space (SAS). The authors are able to simulate geometries of the CSF flow pathways in the brain. However, authors are not familiar with how to compare or represent human dataset in an expected way, particularly when the disease is involved. I have two major concerns and several minor comments.

Response: Thank you very much. We did all of our best for representing the human dataset in a correct way.

Major problems #1:

It seems that there is no validation to show the accuracy of the solution. Only in Table 1, I found a comparison with Experimental values without any reference. In page 19, the authors talked about inlet/outlet BCs and provided few references for that. The Boundary condition strongly depends on the physics of the problem and I doubt if

authors can demonstrate exact solutions or experimental results that maybe comparable to what you have analyzed using FEA software.

Response: We can understand your concern about the importance of boundary conditions (BCs) in computer simulations such as the present study. Accordingly, for better describing these BCs, we have added Table 1 for presenting the BCs of 20 previous studies to compare the previous BCs with BCs “A”, “B”, and “C”, and state the reason for choosing BCs “A”, “B”, and “C”. We tried to mention all of the main sources of previous BCs in Table 1, and we incorporated all of them in the main text of the manuscript;

A- Yellow highlighted phrases in Page 3, Line 19 to Page 4, Line 14.

B- The section “Inlet/outlet boundary conditions of models“ in Pages 19, Line 19 to Page 21, Line 2.

Taken together, we developed previous BCs for choosing the most accurate BC. It should be noted that the geometries of models were similar to each other during the simulations with three different BCs, thus only the differences between BCs were the reason for differences between the results of simulations. We also mentioned this note in the manuscript as follows:

Page 9, Lines 10-12

“It should be noted that these volumes were calculated after 3D modeling of the geometry of the head substructures in CATIA software before analysis. Therefore, these data are independent of the applied BCs "A," "B," and "C".”

Furthermore, we used two indicators for data validation to show the accuracy of the solution and ensuring the correctness of the computer simulation process. We stated these details, and the differences between experimental and simulation data in the section of “Comparison of experimental and biomechanical simulation data“ in Pages 7-9 to demonstrate experimental results are comparable to what we have analyzed using FEA software, and differences between them are in an acceptable range.

The first indicator for data validation was CSF velocities of all healthy subjects and hydrocephalus patients which were measured experimentally using CINE PC-MRI. We compared the maximum (Fig. 3a-c) and phase differences (Fig. 1f,g) of these experimental velocity profiles with the simulation results of CSF velocity profiles. The results of this comparison were acceptable (Page 8, Line 10-16).

The second indicator was CSF pressure in SAS of hydrocephalus patients No. 1-10 which were measured experimentally using ICP-monitoring. The results of the comparison between these pressure values were reported in Table 3 and Page 8, Line 25 to Page 9, Line 3.

It should be noted that one of the main reasons to reject the accuracy of BCs “A” and “B” in comparison with BC “C”, was non-conformity between the experimental data and corresponding simulation data in BCs “A” and “B”. Moreover, based on the results of data validations (especially for CSF pressure) we discussed in the Discussion section about the reason for choosing BC “C” as an accurate BC for simulation of hydrocephalus patients.

Page 11, Lines 3-8:

“There was also a significant nonconformity between the maximum CSF pressure in SAS, which is obtained experimentally from ICP monitoring, and the FSI simulation results of CSF pressure in SAS under BCs “A” and “B.” These nonconformities in BCs “A” and “B” were 36.2% and 14.1%, respectively (Table 3). The biomechanical simulation results of CSF pressure in SAS under BC “C,” however, were close to the maximum CSF pressure in SAS, which is measured experimentally from ICP monitoring, with an error of less than 4.6% (Table 3).”

Major problems #2 and Minor comments or suggestions 3)

Major problems #2:

Table 1 is utterly wrong. At present form, you provide a twisted impression that there are 11 experimental subjects and 1 control (healthy/normal). According to your abstract, there are 8 control subjects. Therefore, your Table 1 should address 1) either all individuals of N=19 or 2) mean, standard deviation per group. For example, Suggested new Table 1.

Case number: Control (n=8) Hydrocephalus (n=11)

Max CSF pressure in SAS 650+/-2 3500+/-10 (mean+/-s.e.)

Max CSF pressure in CA 560+/-2 3100+/-11 (mean+/-s.e.)

&

Minor comments or suggestions:

2) pp. 21 Statistics

You should rewrite. Describe the method. What stat method did you use? You basically compare two groups in your study. Thus, you should specify if you were using “parametric test such as Student t test or independent t test” or “non-

parametric test such as Mann-Whitney U test”. Here is an example of how to write a statistical method that you are expected to use:

“...Statistical analysis

Statistical analyses were performed using SigmaPlot (version 11, Systat Software Inc.) and Prism (GraphPad version 7). Normality of data distribution was tested using the F-test for unequal variance. Normally distributed data were analyzed using Student’s t-test and Tukey’s post-hoc test for pair-wise comparison after ANOVA when comparing two or three groups respectively. Non-normally distributed data were analyzed using the nonparametric Mann-Whitney test and Kruskal-Wallis with Dunnett’s post hoc tests when comparing two and three data groups, respectively. Data were expressed as average \pm standard error of the mean (S.E.) and were considered significant at the $p \leq 0.05$ level....” (source: PMID: 30705305)

Responses for two aforementioned comments: Thank you very much for your specific attention. For considering all of your concerns, the size of Table 1 will be very large. Accordingly, we have added the detailed values of healthy subjects and hydrocephalus patients in the separated Tables 2 and 3, respectively. We also re-arranged the numbers of all Tables in the manuscript.

We have added the standard deviation (SD), standard error (SE) of the mean, coefficient of variation (CV), and confidence interval for all pressures and volumes data of all healthy subjects and hydrocephalus patients in addition to mean values in both Tables 2 and 3 (the right-hand side columns).

We have added SE and CV in Fig. 1a and its caption for aqueductal CSF stroke volume data. We have also added the mean, SD, SE, CV, and confidence interval of the velocities data in Fig. 3a-c (statistical analysis tables). Moreover, for comparison of the CFD and FSI data, we have also added the mean, SD, SE, CV, and confidence interval in Fig. 3g.

Standard errors had been missed in the main text of the previous version of the manuscript, hence, presenting of all values in the main text is corrected according to “Mean \pm SE” in the revised version of the manuscript. We have marked all of them with the yellow highlight in the whole manuscript. Furthermore, we have added some paragraphs in the manuscript to refer to the details of statistical methods and analysis, and their values. It should be noted that we have added Ref. 98 to refer to the below paragraphs.

Page 22, Line 4-13 and 15-17:

“The results of the Shapiro-Wilk test showed that the datasets had a normal distribution. Parametric ANOVA Multiple Comparison was used [98] to compare the CSF pressure, the aqueductal CSF stroke volume, the Reynolds number, the CSF velocity, and the volume between healthy subjects and patients under the three BCs "A," "B," and "C". The homogeneity of the variance test showed that all the variances were equal. Hence, Tukey's post-hoc test was used for pair-wise comparison after ANOVA when comparing the data under the three BCs. Moreover, the normally distributed data led to using Student's t-test after ANOVA to compare both groups of CSF velocities obtained from computer simulation and CINE PC-MRI, and two groups of CSF pressures obtained from computer simulation and ICP-monitoring for assessing data validation. Student's t-test with equal variance was also used to compare the CSF pressure results of CFD and FSI simulations. The test statistics for ANOVA and Student's t-test were T and F, respectively.

.....

Hence, after ensuring the normal distribution of the data, PCC was used to assess the correlation between the maximum CSF pressure and the ventricular system volume under the three BCs. The data were described as mean \pm SE, ...”

Ref 98: Shim JW, Territo PR, Simpson S, Watson JC, Jiang L, Riley AA, McCarthy B, Persohn S, Fulkerson D, Blazer-Yost BL. Hydrocephalus in a rat model of Meckel Gruber syndrome with a TMEM67 mutation. Scientific reports. 2019 Jan 31;9(1):1-7.

Page 8, Lines 4-9:

“The P-values in the Shapiro-Wilk test for all velocity datasets were higher than 0.78, which confirmed the normality of all experimental and FSI simulation data of CSF velocities. The P-values in all Student's t-tests were higher than 0.76. It should be noted that in all Student's t-tests, the P-value of variance equality was higher than 0.64. Moreover, the results of Fig. 3a-c demonstrated that the boundary conditions fall in the 95% confidence interval of the CSF velocity data.”

Page 8, Lines 21-24:

“The P-values in the Shapiro-Wilk test for all CSF pressure datasets were higher than 0.66, which confirmed the normality of all CSF pressure data of ICP monitoring and FSI simulation. The P-values in all Student's t-tests were higher than 0.65. It should be noted that the P-values of variance equality in all Student's t-tests were higher than 0.59. “

Minor comments or suggestions:

1) pp. 8 Line 22 “...Hydrocephalus occurs...”

“(suggestion)”...Hydrocephalus results from imbalance of production and reabsorption in CSF, that involves ventriculomegaly and CSF accumulation in the extra-ventricular space of the brain. Therefore, investigating the geometric and volumetric change of the cranium is of great importance.

Response: These suggested sentences are replaced in the manuscript as follows, furthermore, we have added references 41 and 42 to describe these sentences:

Page 9, Lines 5-7

“Hydrocephalus is caused by an imbalance of production and reabsorption in CSF that involves ventriculomegaly and CSF accumulation in the brain's extra-ventricular space [41,42]. Therefore, investigating the geometric and volumetric changes of the cranium is of great importance...”

Ref 41: McKnight I, Hart C, Park IH, Shim JW. Genes causing congenital hydrocephalus: Their chromosomal characteristics of telomere proximity and DNA compositions. *Experimental neurology*. 2020 Nov 4;113523.

Ref 42: Hochstetler AE, Smith HM, Preston DC, Reed MM, Territo PR, Shim JW, Fulkerson D, Blazer-Yost BL. TRPV4 antagonists ameliorate ventriculomegaly in a rat model of hydrocephalus. *JCI insight*. 2020 Sep 17;5(18).

Minor comments or suggestions:

3) pp. 33 Fig. 1 ...Panels (e) and (f)...

...Panels (f) and (g)...

Response: These letters are corrected in the manuscript as follows:

Page 41, Line 6

“Panels (f) and (g) respectively show the...”

Reviewer #2:

The first comment: We have divided the first comment into three separated comments.

The first comment 1):

1). I think these three boundary condition simulations (A, B, and C) and the cine-flow data collected from 11 hydrocephalus patients and 8 healthy controls could make a useful contribution--and they appear to be novel. However, this paper is probably too long,

Response: Thank you very much. The descriptions of all details of BCs (A, B, and C), and solving methods (CFD and FSI) led to this long paper. Approximately, 40% of the word counts are dedicated to the Method section. For fully understanding the readers about the necessity of using these BCs and solving methods, and describing the actual novelty and necessity of the present study, writing these descriptions were somehow inevitable.

But according to your comment, we have re-written some parts of the manuscript to decrease the word counts. Moreover, we have removed some sentences in the original manuscript as follows:

Page 6, Lines 11, 12.

Page 8, Lines 13, 14.

Page 11, Line 10.

Page 14, Lines 13-16.

Page 15, Line 22, 23.

Decreasing the word counts of Abstract section.

The first comment 2):

2) it needs to add some sort of statistical analysis (e.g., do the boundary conditions fall in the 95% confidence interval of the cine-flow data?),

Response: According to the concern of the respected reviewer about statistical analysis, we have reported the statistical methods, analysis, and detailed values, for cine-flow data and also for all of the other data.

We have added the standard deviation (SD), standard error (SE) of the mean, coefficient of variation (CV), and confidence interval for all pressures and volumes data of all healthy subjects

and hydrocephalus patients in addition to mean values in both Tables 2 and 3 (the right-hand side columns).

We have added SE and CV in Fig. 1a and its caption for aqueductal CSF stroke volume data. We have also added the mean, SD, SE, CV, and confidence interval of the velocities data in Fig. 3a-c (statistical analysis tables). Moreover, for comparison of the CFD and FSI data, we have also added the mean, SD, SE, CV, and confidence interval in Fig. 3g.

Standard errors had been missed in the main text of the previous version of the manuscript, hence, presenting of all values in the main text is corrected according to “Mean±SE” in the revised version of the manuscript. We have marked all of them with the yellow highlight in the whole manuscript. Furthermore, we have added some paragraphs in the manuscript to refer to the details of statistical methods and analysis, and their values.

Page 22, Line 4-13 and 15-17:

“The results of the Shapiro-Wilk test showed that the datasets had a normal distribution. Parametric ANOVA Multiple Comparison was used [98] to compare the CSF pressure, the aqueductal CSF stroke volume, the Reynolds number, the CSF velocity, and the volume between healthy subjects and patients under the three BCs "A," "B," and "C". The homogeneity of the variance test showed that all the variances were equal. Hence, Tukey's post-hoc test was used for pair-wise comparison after ANOVA when comparing the data under the three BCs. Moreover, the normally distributed data led to using Student's t-test after ANOVA to compare both groups of CSF velocities obtained from computer simulation and CINE PC-MRI, and two groups of CSF pressures obtained from computer simulation and ICP-monitoring for assessing data validation. Student's t-test with equal variance was also used to compare the CSF pressure results of CFD and FSI simulations. The test statistics for ANOVA and Student's t-test were T and F, respectively.

.....

Hence, after ensuring the normal distribution of the data, PCC was used to assess the correlation between the maximum CSF pressure and the ventricular system volume under the three BCs. The data were described as mean ± SE, ...”

Page 8, Lines 4-9:

“The P-values in the Shapiro-Wilk test for all velocity datasets were higher than 0.78, which confirmed the normality of all experimental and FSI simulation data of CSF velocities. The P-values in all Student's t-tests were higher than 0.76. It should be noted that in all Student's t-tests, the P-value of variance equality was higher than 0.64. Moreover, the results of Fig. 3a-c demonstrated that the boundary conditions fall in the 95% confidence interval of the CSF velocity data.”

Page 8, Lines 21-24:

“The P-values in the Shapiro-Wilk test for all CSF pressure datasets were higher than 0.66, which confirmed the normality of all CSF pressure data of ICP monitoring and FSI simulation. The P-values in all Student’s t-tests were higher than 0.65. It should be noted that the P-values of variance equality in all Student’s t-tests were higher than 0.59. “

The first comment 3):

3) and it would be helpful to readers if the authors framed better in the Discussion section or Conclusion how these results extend our knowledge concerning hydrocephalus.

Response: The neurosurgeons and neurophysicians use neuroimages, as well as the history and symptoms of hydrocephalus patients as primary tools for diagnosing and treating the patients. However, making decisions only based on these tools is highly difficult. For instance, based on these diagnostic tools, some hydrocephalus types are misdiagnosed with Parkinson’s or Alzheimer’s disease while hydrocephalus is reversible and its distinct diagnosis is highly valuable. On the other hand, the results of the present study demonstrated that the maximum CSF pressure is the most significant parameter for the distinct diagnosis of hydrocephalus. Measuring this parameter is not possible using current experimental noninvasive methods while it is calculatable very accurate using biomechanical simulation process which is suggested in the present study noninvasively. In ref 48 and 33 we had discussed about the reason of the importance of CSF pressure in diagnosing and treating hydrocephalus patients, respectively. Accordingly, we have added a brief paragraph (because mentioning all the above details, increases the wordcounts significantly) at the end of the Discussion section as follows:

Page 13, Line 16-20.

“As one of the most significant parameters in the evaluation of hydrocephalus patients, measuring the maximum CSF pressure is not possible using current experimental noninvasive methods. The present noninvasive biomechanical simulations are useful for accurately measuring the CSF hydrodynamic parameters, such as CSF pressure changes, using the suggested BC and solving method, which help diagnose [48] and treat [33] hydrocephalus patients.”

The second comment:

This paper emphasized CSF pressure in the cerebral aqueduct (CA) and the subarachnoid space (SAS) in terms of characterizing hydrocephalus. However, Park et al. (2010, Journal of Neurosurgery: Pediatrics) used an animal (dog) model in which pre- and post-fourth-ventricle obstruction was used to assess the cardiac pulsation absorbance (CPA) index. They found that the CPA index went down significantly after chronic hydrocephalus was induced in these animals. This suggested that a major part of the problem with hydrocephalus is that the input flow to micro-vessels in the brain is disrupted. How would the present model incorporate the vascular changes in hydrocephalus?

Response: Due to the very challenging nature of this comment, the majority of time for responding to all reviewers has been dedicated to this comment.

Our input flows were obtained based on the in vivo blood flow of all subjects which were measured using CINE PC-MRI in the basilar artery (Page 16, Lines 16-19, and Page 20, Lines 7-10, 21, and Page 21, Lines 1,2). So, the effects of vascular changes and cardiac pulsation are considered in the inlet flow of each subject using the blood flow diagram in the basilar artery. Although we have used the in vivo data of the blood flow for defining the inlet flow, our main concentration was about accurate BC and solving method based on CSF pressures results, not exactly cardiac pulsation absorbance ... But your concern was really important and controversial for us. Hence, we reassessed all of our data sources of blood flow profiles for finding the relationship between amplitude and/or other characters of blood flow profile with the changes in the ICP diagram. Even we also reassess the relationships of blood flow profile with ICP and also with ICC in our previous paper [doi: 10.1371/journal.pone.0196216] that I had followed-up hydrocephalus patients for 2.5 years. Although we reassessed all of these data in a very time-consuming process, unfortunately, they were not a meaningful relationship between the aforementioned parameters.

First, we thought maybe the locations that we have calculated the CSF pressure diagrams (CA and SAS) were not correct, and maybe according to the study by Park et al., there is only a relationship between blood flow and CSF pressure profiles in the fourth ventricle. Accordingly, we have recalculated CSF pressure for the values of CSF pressure in the fourth ventricle (below Tables). Unfortunately, the general trend of the CSF pressure changes in the fourth ventricle was not

different with CA or SAS, and there was not also any meaningful relationship between blood flow profile in the basilar artery and CSF pressure in the fourth ventricle and this did not work for us either.

Patients											
Patients number	1	2	3	4	5	6	7	8	9	10	11
CSF pressure in the fourth ventricle under the BC “C”	2543	2632	2410	2655	2409	2487	2394	2382	2398	2664	3635
	.8	.4	.1	.2	.3	.1	.7	.7	.4	.9	.9

Normal subjects									
Healthy subjects’ number	1	2	3	4	5	6	7	8	
CSF pressure in the fourth ventricle under the BC “C”	503.7	506.3	509.2	994.8	459.2	486.5	508.2	453.1	

It is worth mentioning that we can sure about the correctness of the above Tables. Since if you look at Table IV in Ref 27, the differences between pressure values in lateral ventricles and SAS were also very small as similar to our study, which means the above tables in comparison to Tables 2 and 3 are correct. It should be noted that we did not report the above Tables in the manuscript and we report these results only for the respected reviewer.

Dear Reviewer, all of our efforts for finding the relationships were ineffective. We think these reasons could be involved in this happening:

- 1- Our inlet data came from the basilar artery, and maybe the blood flow profile in this artery cannot reflect the effects of the input flow to micro-vessels that Park et al. had mentioned.
- 2- Maybe this can be related to micro-mechanism of the CSF secretion and the defining of the location of the production and/or absorptions of CSF in the computerized models of hydrocephalus patients. Because approximately in all of the previous simulation studies, the location for applying the inlet flow was assumed in the lateral ventricles (Page 20, Lines 11-20). While in microscopic approach there is the CSF secretions in all ventricular system especially in lateral and fourth ventricles and also in choroid plexuses of all CSF circulation system [48] that they could not consider in the computer simulations.

It should be noted that despite the aforementioned concerns, we used two indicators for validation of our results and ensuring the results and correctness of our computer simulation process. One of

them was CSF velocity which was measured using CINE PC-MRI, and another one was CSF pressure in SAS which was measured using ICP-monitoring. The results of data validation for velocity and pressure data with the computer simulation results were really acceptable (Page 8, Lines 10-16) and (Page 8, Line 25 to Page 9, Line 3). Furthermore, in addition to the acceptable results of data validations, our results in comparison to the results of previous studies were also acceptable (Page 6, Lines 8, 9-11). This means maybe our model could not reflect the microscopic complexities such as the effects of the input flow to micro-vessels, but according to data validations and compare our results with the results of previous studies, our results are enough correct for supporting the goals of the present study.

According to the aforementioned notes, we have added a brief paragraph in the manuscript as follow:

Page 14, Lines 13-19:

“Park et al. revealed that the cardiac pulsation absorbance went down significantly after chronic hydrocephalus in animal subjects [62]. They also found that a significant part of this problem with hydrocephalus was the disrupted input flow to micro-vessels in the brain. Regarding the approach and goals of the present simulation study, this phenomenon could not make a proper assessment. Hence, future studies are suggested to consider this effect in the inlet BC and assess changes in cardiac pulsation absorbance in the simulation results using a microscopic approach to utilize this parameter as complementary metrics in the evaluation of hydrocephalus patients. Future studies are also recommended to evaluate the relationship between BCs and symptom severities in larger sample sizes.”

Ref 62: Park EH, Dombrowski S, Luciano M, Zurakowski D, Madsen JR. Alterations of pulsation absorber characteristics in experimental hydrocephalus. Journal of Neurosurgery: Pediatrics. 2010 Aug 1;6(2):159-70.

The third comment:

Finally, did the boundary conditions (A, B, or C) or cine-flow data correlate with symptom severity in hydrocephalus patients?

Response: First, according to your attention about boundary conditions, we improved descriptions about BCs in the manuscript by adding Table 1 and its corresponding notes in the manuscript as follows:

A- Yellow highlighted phrases in Page 3, Line 19 to Page 4, Line 15.

B- The section “Inlet/outlet boundary conditions of models“ in Pages 19, Line 19 to Page 21, Line 2.

Dear reviewer, I (corresponding author) had published three papers [[10.1016/j.jocn.2016.09.012](https://doi.org/10.1016/j.jocn.2016.09.012); [10.1038/s41598-020-72961-0](https://doi.org/10.1038/s41598-020-72961-0); [10.1016/j.wneu.2018.05.108](https://doi.org/10.1016/j.wneu.2018.05.108)] with large number of patients. In those papers, I had assessed the relationship of symptom severity of patients with the CSF dynamic changes which were calculated using CINE PC-MRI and computer simulations.

But the number of subjects in the present study is not enough for discussing about clinical symptoms of patients and their severity. We think if we had reported any discussions about clinical symptoms, without any doubt, we could not defense to our findings and we could not respond to possible reviewers’ questions because of a small number of subjects in comparison to the standard of a clinical study. Hence, we did not involve the symptoms in this article. In the present study, we only attempted to find an optimal BC and solving method for accurate computerized biomechanical simulation of hydrocephalus and we used two indicators (CSF velocity which was measured by CINE PC-MRI, and CSF pressure which was measured by ICP-monitoring) for data validation.

It should be noted that the number of subjects in the present study is much higher than the majority of previous computer simulation studies. Since computer simulation is a very time-consuming process and providing the results of the present study lasted more than 3.5 years. If these simulations were only CFD, this time had a significant reduction. But FSI simulations of the present study were extremely time-consuming and now we really could not add more subjects to increase the number of subjects for assessing the clinical symptoms and adding a discussion about the relationship of BCs and CINE PC-MRI results with symptom severities. Hence, we would be appreciated if you consider our concern.

But we have added a sentence in the manuscript as follows to compensate for this shortcoming in future studies:

Page 14, Lines 19, 20:

“Future studies are also recommended to evaluate the relationship between BCs and symptom severities in larger sample sizes.”

Reviewer #3

Comment:

1- The abstract is quite long and densely written. I propose to shorten it and focus on the main message, e.g. FSI model advantages. I would pick up boundary conditions in the discussion, but shorten its significance in the abstract, because the write up is otherwise too technical.

Response: According to the limitation in the wordcounts of the Abstract section in the author guidelines of this journal, we have rewritten the Abstract section to consider your concern.

Comment:

2- Intracranial Pressure (ICP) and CSF pressure — it would improve the clarity of the study if the authors could elaborate more on the differences between the ICP and CSF pressure. How the ICP was obtained (computed) in the models?

Response: We explained in the below sentences:

Page 6, Lines 13-21:

“As the absolute value of ICP is not directly measurable using MRI, the analysis of intracranial dynamics cannot be performed entirely based on imaging. While having information on the absolute ICP value could be essential for assessing patients' clinical conditions. It is impossible to calculate the absolute value of ICP by computational fluid mechanics, as only the pressure gradient appears in fluid motion equations. In biphasic models, pressure acts as the fluid phase's pressure, but its level is undetermined. Hence, we did not calculate absolute ICP. In this study, as in many previous studies, CSF pressure calculated by biomechanical simulation in the pathway of CSF circulation is called CSF pressure [26,30,32,33], and CSF pressure in the upper convexity of the brain in SAS and CA is called CSF pressure in SAS and CSF pressure in CA, respectively.”

Page 8, Lines 17-19:

“Second, the SAS's CSF pressure values were experimentally measured in 10 of the 11 patients by continuous monitoring of ICP using a reliable sensor introduced 1–2 cm through a minimal opening and a small burr hole [40].”

Comment:

3-The description of the methods of computer simulation is not comprehensive. The information about mesh information is incomplete and the authors should add more details about them.

Response: We have explained in the below sentences:

Page 16, Lines 23-26:

“In an FSI problem, the computational grid of the fluid model deforms. Hence, in the ALE formulation, velocity relative to mesh velocity is used in the momentum equation's convective term. It means that velocity at the faces of a moving and deforming control volume is the relative velocity concerning the faces of the control volume (or the computational grid) [67].”

Page 21, Lines 5 and 6:

“An implicit Euler scheme was used to refine the grids successively with a time step of 0.01 through transient analysis.”

Comment:

4- Why do you neglect the skull bone in your 3D geometric model. What will be the effect of this on your results?

Response: This concern is really important and there were many studies concerning the effect of the skull bone. In many previous papers that the main goal of them was related to head impact, the effects of skull bone were considerable. But my previous paper proof that the power of CSF flow is not considered in front of the rigidity of the skull. And the effect of considering the skull bone in the study with a focus on CSF dynamics is negligible [33]. We added this ref in the manuscript.

Page 17, Lines 22-24:

“It has been demonstrated in Gholampour's study that the deformation of SAS's outer surfaces in all directions is negligible and that defining the FSI boundary for SAS's outer surfaces has almost no considerable effect on results [33].”

Comment:

5- The FSI method which used in this study was one-way or two-way? Do you think the other method which you don't use in your study, will have a significant impact on your results? Why?

Response: The present problem was solved under the two-way FSI with strong coupling. This means that the solutions for all the physics be synchronized at every time step. The main advantage of this approach is that optimized existing solvers can be reused and coupled. If the other types of FSI simulation such as one-way FSI or two-way FSI with weak coupling are used for the calculation, the errors will be increased. We have added this in Page 16, Line 21:

“Simulations were performed using a fully coupled FSI (two-way FSI with strong coupling) model,...”

Comment:

6- Do you re-calculate all models of all samples using CFD simulation as similar to FSI? Clarify this in the main text. Because it is not clear that Fig. 3g is the result of which comparison. Is it the result of the comparison of all samples before and after treatment process?

Response: As mentioned in the method section, we did not follow-up the subjects.

Page 12, Line 25 to Page 13, Line 3:

“The no-slip BCs governing Equation (6) were assumed for all SAS's inner and outer surfaces. The inlet and outlet flow rate functions were completely similar to those in BC "C," and the outlets' locations were defined as those in BC "C." However, the CSF inlet location was defined after the foramina of Luschka and Magendie (the inlet of the SAS area). The problem was then analyzed using the Navier-Stokes equation [Eq. (1)].”

Comment:

7- Add the names of some types of hydrocephalus which CFD method can be a better method for simulating of them. Also, for FSI method. Clearly add the samples for internal and external hydrocephalus in your discussion section.

Response: We have added those names in the manuscript as follows:

Page 13, Lines 10-13:

“Thus, the CFD method can be used to calculate CSF pressure in SAS for external hydrocephalus, such as Chiari malformation and pseudotumor cerebri, which occurs due to a cause outside the ventricular system. In contrast, in internal hydrocephalus diseases, such as aqueductal stenosis, mesencephalic tumor, and aqueductal web,...”

REVIEWERS' COMMENTS:

Reviewer #1 (Remarks to the Author):

Authors have revised the manuscript as anticipated by addressing all questions that I have raised. Therefore, I have no further comments or objection.

Reviewer #2 (Remarks to the Author):

I now believe that the authors have addressed the Reviewers' earlier concerns in a reasonable manner. Therefore, I now recommend publication of this paper.

Reviewer #3 (Remarks to the Author):

Accept